# Inhibition of RANK signaling in breast cancer induces an anti-tumor immune response orchestrated by CD8+ T cells

Clara Gómez-Aleza ⬥ et al.#

Most breast cancers exhibit low immune infiltration and are unresponsive to immunotherapy. We hypothesized that inhibition of the receptor activator of nuclear factor-κB (RANK) signaling pathway may enhance immune activation. Here we report that loss of RANK signaling in mouse tumor cells increases leukocytes, lymphocytes, and CD8+ T cells, and reduces macrophage and neutrophil infiltration. CD8+ T cells mediate the attenuated tumor phenotype observed upon RANK loss, whereas neutrophils, supported by RANK-expressing tumor cells, induce immunosuppression. RANKL inhibition increases the anti-tumor effect of immunotherapies in breast cancer through a tumor cell mediated effect. Comparably, pre-operative single-agent denosumab in premenopausal early-stage breast cancer patients from the Phase-II D-BEYOND clinical trial (NCT01864798) is well tolerated, inhibits RANK pathway and increases tumor infiltrating lymphocytes and CD8+ T cells. Higher RANK signaling activation in tumors and serum RANKL levels at baseline predict these immune-modulatory effects. No changes in tumor cell proliferation (primary endpoint) or other secondary endpoints are observed. Overall, our preclinical and clinical findings reveal that tumor cells exploit RANK pathway as a mechanism to evade immune surveillance and support the use of RANK pathway inhibitors to prime luminal breast cancer for immunotherapy.

---

#A list of authors and their affiliations appears at the end of the paper.

Breast cancer (BC) in young women has a unique biology and is associated with poor prognosis. Previous results support a role for the receptor activator of nuclear factor-κB (RANK) signaling pathway in these tumors[1]. RANK pathway plays a crucial role in bone remodeling and mammary gland development[2,3], acting as a paracrine mediator of progesterone for the expansion of mammary stem/progenitor cells, and mediates the early steps of progesterone-driven mammary tumorigenesis[4–7]. Denosumab is a human monoclonal antibody against RANK ligand (RL), approved for the prevention of skeletal morbidity associated with metastatic bone disease and the management of treatment-induced bone loss in early postmenopausal BC. Preclinical data reinforce the potential role of RL inhibitors such as denosumab in BC prevention[4,5,8,9] and treatment due to its ability to reduce recurrence and metastasis[10]. We previously found that RANK loss in the oncogene-driven mammary tumor model MMTV-PyMT (PyMT) significantly reduced tumor incidence and lung metastases[10]. Tumor cells lacking RANK showed delayed tumor onset and a reduced ability to initiate tumors and metastasis. Pharmacological inhibition of RL also reduced tumor-initiating ability and led to the lactogenic differentiation of tumor cells[10].

RANK and RL are expressed in a wide variety of immune cells[11] and are involved in various immune processes, including lymph node development[12], the activation of dendritic cells, monocytes and T cells, and the establishment of central and peripheral tolerance[11–19]. Thus, RANK pathway regulates innate and adaptive immune responses, and may promote or suppress immunity, depending on the context.

Tumor cells develop several strategies to evade immune surveillance: reducing infiltration by cytotoxic T lymphocytes or natural killer (NK) cells and increasing recruitment of immunosuppressive cells, such as regulatory T cells (Tregs) and different myeloid populations, such as tumor-associated macrophages (TAMs) and tumor-associated neutrophils (TANs)[20]. Immune-checkpoint inhibitors (mainly antibodies against cytotoxic T-lymphocyte-associated protein 4 (CTLA4) and programmed cell death protein-1 (PD-1) and its ligand (PD-L1)) have emerged as potent therapies against some solid tumors such as melanoma and advanced non-small cell lung cancer (NSCLC)[21,22]. Nevertheless, in BC the efficacy of immunotherapy remains limited even after the inclusion of radiotherapy or chemotherapy[23], in particular in the immune "cold" luminal tumors.

Here, exploiting complementary genetic and pharmacological approaches in the PyMT tumor model[24], we investigate the effects of RANK pathway inhibition on mammary tumor immune surveillance. RANK and RL expression patterns in PyMT tumors resemble those found in human breast adenocarcinomas, with RANK being expressed in tumor cells and myeloid cells, and RL in tumor-infiltrating lymphocytes (TILs)[4,10,25,26]. RANK deletion in tumor cells, but not in myeloid cells, leads to an increase in immune, lymphocyte, and CD8+ T-lymphocyte infiltration, and a reduction in the infiltration of myeloid cells. TANs and CD8+ T lymphocytes modulate the anti-tumor immune response driven by loss of RANK expression in tumor cells. Systemic RL inhibition also increases CD8+ T-cell infiltration and reinforces the anti-tumor benefits of checkpoint inhibitors in RANK-positive tumors. Importantly, the immune-modulatory effect of RANK signaling is confirmed in the D-BEYOND (denosumab, a RANK-ligand (RANKL) inhibitor and its Biological Effects in YOuNg premenopausal women Diagnosed with early breast cancer) clinical trial (NCT01864798), a prospective, pre-operative study evaluating denosumab and its biological effects in premenopausal early-stage BC. Two courses of denosumab induce an increase in TILs and CD8+ T-cell infiltration. Increased activation of RANK signaling pathway in the tumors and circulating serum RL at baseline are identified as predictive biomarkers for the denosumab-driven increase in TILs. Together, these results demonstrate the key role of RANK pathway in the tumor-immune crosstalk and support the use of RL inhibitors, such as denosumab, for enhancing the immune response in poorly immunogenic luminal BC.

## Results

**Loss of RANK in tumor cells leads to increased lymphocyte infiltration.** We hypothesized that, beyond its tumor cell-intrinsic effects[10], inhibition of RANK signaling pathway may enhance immune activation in BC. To test this hypothesis, we undertook genetic approaches using the PyMT luminal tumor mouse model. First, we tested whether loss of RANK signaling in myeloid cells could induce changes in immune infiltration, by using LysM-cre/RANK^{flox/flox} mice. Expression of Cre driven by LysM deletes RANK in the myeloid compartment (RANK MC$^{-/-}$)[27]. As expected, lower levels of *Rank* mRNA were found in peritoneal macrophages from RANK MC$^{-/-}$ mice (Fig. 1a). PyMT RANK$^{+/+}$ (RANK$^{+/+}$) tumors were orthotopically transplanted in RANK MC$^{-/-}$ mice and corresponding controls (RANK MC$^{+/+}$) (Fig. 1a). Analyses of the tumor immune infiltrates revealed no changes in immune infiltration, leukocytes (CD45$^+$), lymphocytes (CD11b$^-$ within CD45$^+$), TAMs (F4/80$^+$CD11b$^+$ within CD45$^+$), or TANs (Ly6G$^+$ CD11b$^+$ within CD45$^+$) between genotypes (Fig. 1b and Supplementary Fig. 1a, b). The frequencies of infiltrating CD8$^+$ T cells (CD11b$^-$ CD3$^+$ CD8$^+$), CD4$^+$ T cells (CD11b$^-$ CD3$^+$ CD8$^-$), and the CD4/CD8 ratio were also similar in RANK$^{+/+}$ tumors growing in RANK MC$^{-/-}$ or RANK MC$^{+/+}$ mice (Supplementary Fig. 1a, b).

We next tested whether RANK loss exclusively in tumor cells could alter tumor immune infiltration: tumors derived from PyMT/RANK$^{-/-}$ mice (RANK$^{-/-}$ tumors) were orthotopically transplanted in syngeneic C57Bl6 mice and compared with RANK$^{+/+}$ tumor transplants. RANK$^{-/-}$ tumors showed greater infiltration by leukocytes, lymphocytes, and CD8$^+$ T cells compared with RANK$^{+/+}$ tumors of similar size (Supplementary Fig. 1a, c). Together, these results demonstrate that loss of RANK in tumor cells, but not in myeloid cells, induces an increase in tumor-immune infiltrates, TILs, and CD8$^+$ T cells.

**T cells mediate the longer tumor latency of RANK$^{-/-}$ tumors.** The increase in TILs observed after loss of RANK in tumor cells, prompted us to investigate the functional contribution of this immune population. To this end, RANK$^{+/+}$ and RANK$^{-/-}$ tumor cells were transplanted in parallel in syngeneic mice and in T-cell-deficient *Fox1$^{nu}$* mice (Fig. 1c). We had previously demonstrated that, compared with RANK$^{+/+}$, RANK$^{-/-}$ tumor cells display prolonged latency to tumor formation, increased apoptosis, and a lower frequency of tumor-initiating cells when transplanted in syngeneic mice[10].

Strikingly, when transplanted in T-cell-deficient *Foxn1$^{nu}$* mice, no differences in latency to tumor onset were observed between RANK$^{+/+}$ and RANK$^{-/-}$ tumor transplants, whereas the same tumors transplanted in syngeneic C57BL/6 mice corroborated previous results (Fig. 1d and Supplementary Fig. 2a)[10]. In addition, limiting dilution assays in *Foxn1$^{nu}$* mice showed no differences in the ability of RANK$^{+/+}$ and RANK$^{-/-}$ tumor cells to initiate tumors (Fig. 1e). Further characterization of the tumors revealed that RANK$^{-/-}$ tumor transplants growing in syngeneic hosts contained more apoptotic and necrotic cells than did their RANK$^{+/+}$ counterparts (Supplementary Fig. 2b), corroborating previous findings[10]. However, the frequency of apoptotic cells was similar in RANK$^{-/-}$ and RANK$^{+/+}$ tumor cells growing in

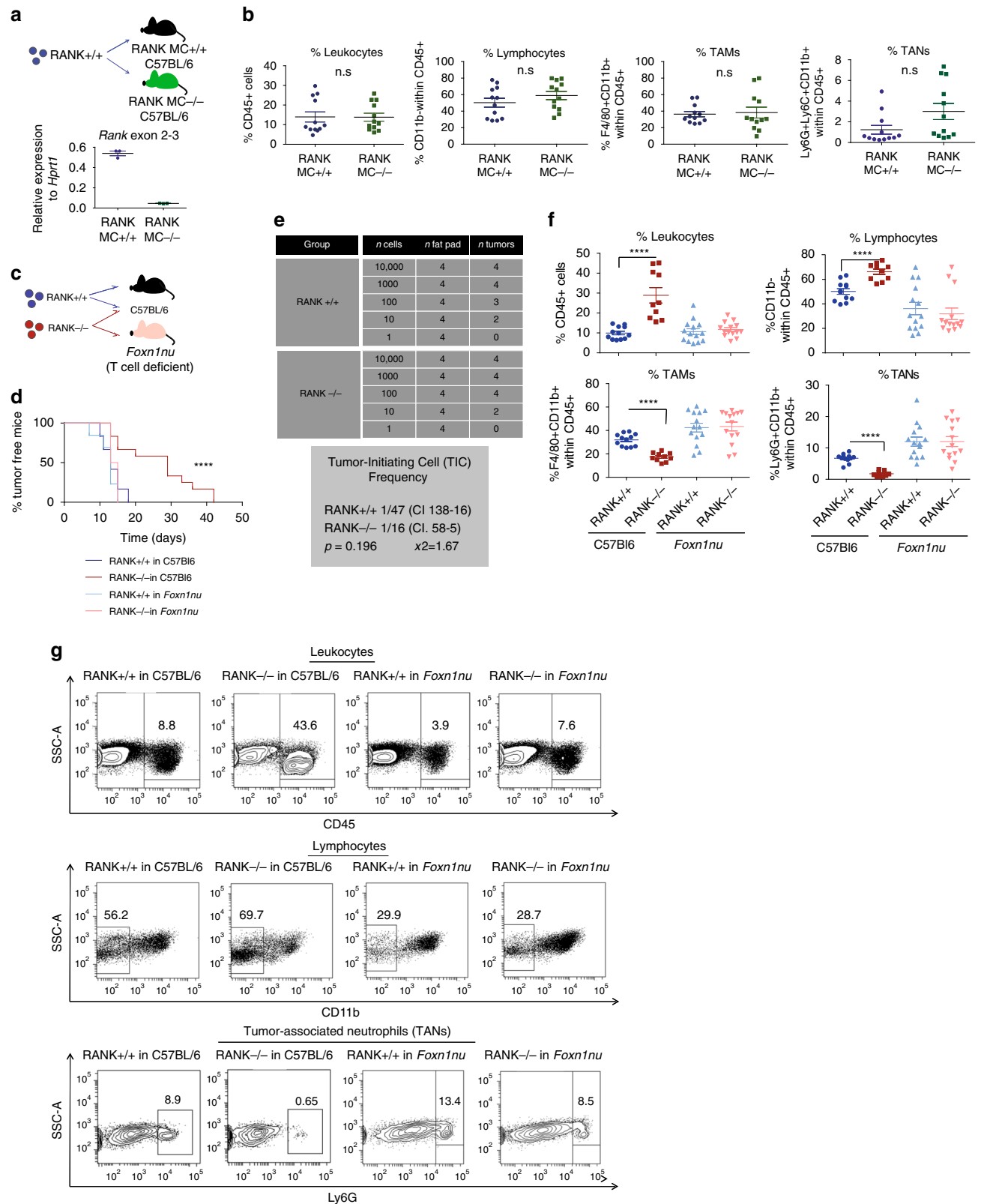

*Foxn1nu* mice. Differences in late apoptosis/necrosis (7AAD[+]/Annexin V[+] cells) between RANK[+/+] and RANK[−/−] tumor cells were observed in both syngeneic and *Foxn1nu* recipients, but were less marked in T-cell-deficient mice (Supplementary Fig. 2b). These observations suggest that the increased tumor cell death rate in the absence of RANK is due to a combination of tumor

cell-intrinsic and T-cell-mediated effects, whereas T cells are responsible for the delayed tumor onset and the reduced tumor-initiating ability of RANK-null tumor cells.

Analyses of RANK[+/+] and RANK[−/−] tumors confirmed the higher frequency of leukocytes and the enrichment in TILs in RANK[−/−] compared with RANK[+/+] tumors (Fig. 1f, g and

**Fig. 1 Loss of RANK in tumor cells, but not in myeloid cells, leads to increased TIL frequency, and T cells drive the delayed tumor formation and the reduced tumor-initiating ability of RANK-null tumor cells. a** Top panel: injection scheme showing the implantation of PyMT RANK$^{+/+}$ (RANK$^{+/+}$) tumors in LysM-Cre RANK$^{fl/fl}$ mice (RANK MC$^{-/-}$) and WT (RANK MC$^{+/+}$) (C57BL/6). Bottom panel: *Rank* mRNA expression levels relative to *Hprt1* in peritoneal macrophages of RANK MC$^{-/-}$ and RANK MC$^{+/+}$ mice ($n = 3$). Mean ± SEM is shown. **b** Graphs showing the percentages of tumor-infiltrating leukocytes (CD45$^+$), lymphocytes (CD11b$^-$ within CD45$^+$), tumor-associated macrophages (TAMs) (F4/80$^+$CD11b$^+$ within CD45$^+$) and tumor-associated neutrophils (TANs) (Ly6G$^+$Ly6C$^-$CD11b$^+$ within CD45$^+$) in RANK$^{+/+}$ tumor transplants in RANK MC$^{-/-}$ and RANK MC$^{+/+}$ mice ($n = 12$ tumors). Mean, SEM shown. *t*-test and *p*-values were calculated. **c** Injection scheme showing the implantation of PyMT RANK$^{+/+}$ and PyMT RANK$^{-/-}$ tumors in C57BL/6 WT animals and *Foxn1$^{nu}$* mice. **d** Kinetics of palpable tumor onset (left) after tumor transplantation of RANK$^{+/+}$ and RANK$^{-/-}$ tumor cells in syngeneic C57BL/6 ($n = 6$) and *Foxn1$^{nu}$* mice ($n = 7$). Log-rank test performed with two-tailed *p*-value (****$p = 0.005$). One representative experiment out of two is shown. **e** Tumor-initiating frequencies as calculated by ELDA. Cells isolated from RANK$^{+/+}$ and RANK$^{-/-}$ tumors were injected in *Foxn1$^{nu}$* mice in limiting dilutions. WEHI's online ELDA-software (http://bioinf.wehi.edu.au/software/elda/) was used to calculate the $\chi^2$-values with 95% confidence interval. **f** Graphs showing the percentages tumor-infiltrating leukocytes (CD45$^+$; ****$p < 0.0001$), lymphocytes (CD11b$^-$ within CD45$^+$; ****$p < 0.0001$), TAMs (F4/80$^+$CD11b$^+$ within CD45$^+$; ****$p < 0.0001$), TANs (Ly6G$^+$CD11b$^+$ within CD45$^+$; ****$p < 0.0001$) in RANK$^{+/+}$ or RANK$^{-/-}$ tumor transplants in syngeneic C57BL/6 and *Foxn1$^{nu}$* mice ($n = 12$ RANK$^{+/+}$ tumors, $n = 10$ RANK$^{-/-}$ tumors in C57BL/6 hosts; $n = 14$ RANK$^{+/+}$ or RANK$^{-/-}$ tumors in *Foxn1$^{nu}$* hosts). Tumors were analyzed at endpoint (>0.2 cm$^2$). Mean, SEM and *t*-test two-tailed *p*-values are shown. Two representative primary tumors were used in these experiments. **g** Representative dot blots of leukocytes (CD45$^+$) gated in live cells (7AAD$^-$) and lymphocytes (CD11b$^-$) gated on CD45$^+$.

Supplementary Fig. 1c). In contrast, the relative frequency of TAMs and TANs was higher in RANK$^{+/+}$ than in RANK$^{-/-}$ tumors (Fig. 1f, g and Supplementary Fig. 1c). These differences were no longer observed in *Foxn1$^{nu}$* transplants (Fig. 1f, g).

To rule out the possibility that immune cells transplanted along with tumor cells were responsible for the observed changes, the CD45$^-$ population (tumor cell-enriched) was sorted and transplanted into syngeneic hosts. The longer tumor latency observed in RANK$^{-/-}$ was exacerbated when sorted CD45$^-$ cells were injected, compared with whole tumor transplants (Supplementary Fig. 2c). Accordingly, differences in immune infiltration were also observed between tumors derived from sorted CD45$^-$ RANK$^{+/+}$ and CD45$^-$ RANK$^{-/-}$ cells and those derived from whole tumor transplants (Supplementary Fig. 2d).

To confirm that our findings are not affected by differences other than RANK status between RANK$^{+/+}$ and RANK$^{-/-}$ tumors, we infected PyMT/RANK$^{flox/flox}$ tumors with pLVX-Cre-IRES-zsGreen or control lentivirus. Infected tumor populations were fluorescence-activated cell sorting (FACS)-sorted and orthotopically transplanted into C57BL/6 mice. RANK depletion was confirmed by reverse transcription PCR (RT-PCR) and immunohistochemistry (IHC) (Supplementary Fig. 2e). RANK-depleted tumors showed lower tumor growth rate (Supplementary Fig. 2f) and greater infiltration of leukocytes, lymphocytes, and T cells (CD3$^+$ CD11b$^-$CD45$^+$), corroborating previous findings (Supplementary Fig. 2g). CD8$^+$ T cells were more abundant and TANs were reduced in RANK-depleted tumors, although the differences were not significant (Supplementary Fig. 2g). Thus, RANK loss in tumor cells leads to a significant increase in TILs.

Together, these results demonstrate that RANK loss in tumor cells leads to a significant increase in TILs that restrict RANK$^{-/-}$ tumor cell growth. Reciprocally, they indicate that RANK expression in tumor cells induces an immunosuppressive microenvironment enriched in TAMs and TANs, allowing tumor cells to escape T-cell immune surveillance.

**CD8$^+$ T cell depletion rescues the delay in tumor onset of RANK$^{-/-}$ tumors.** Further characterization of TIL subsets from syngeneic transplants (Supplementary Fig. 1a), revealed a significant increase in the percentage of CD3$^+$ T lymphocytes and CD8$^+$ T cells in RANK$^{-/-}$ tumors and a lower CD4$^+$/CD8$^+$ ratio in RANK$^{-/-}$ compared with the RANK$^{+/+}$ tumors (Fig. 2a). There were no significant differences between the two groups in the frequencies of NK cells (NK1.1$^+$ CD3$^-$), B cells (CD19$^+$ CD3$^-$CD11b$^-$), or levels of interferon-γ (IFNγ)

production by tumor-infiltrating CD4$^+$ and CD8$^+$ T cells (Supplementary Fig. 3a). However, TAMs that infiltrated RANK$^{-/-}$ tumors expressed higher levels of IL-12/IL23, indicative of an anti-tumor M1 response (Supplementary Fig. 3a). Increased CD3$^+$ T-lymphocyte and CD8$^+$ T-cell tumor infiltration in RANK$^{-/-}$ tumors compared with RANK$^{+/+}$ was confirmed by IHC (Fig. 2b, c) and the mRNA levels of the cytotoxicity markers, namely *Ifnγ* and perforin (*Prf1*) were higher in RANK$^{-/-}$ tumors (Fig. 2d). Gene expression analysis comparing sorted CD45$^-$ cells (tumor cell-enriched) isolated from RANK$^{+/+}$ vs. RANK$^{-/-}$ tumor transplants revealed 604 differentially expressed genes (Supplementary Data 1). Gene Ontology (GO) and Generally Applicable Gene Set Enrichment (GAGE) analyses revealed that RANK$^{-/-}$ tumor cells overexpressed a subset of genes related to the "intrinsic apoptotic signaling pathway," "antigen processing and presentation," and "positive regulation of T-cell-mediated cytotoxicity" (Supplementary Data 2–4). Similar frequencies of CD3$^+$, CD4$^+$, and CD8$^+$ T cells were found in draining lymph nodes from RANK$^{+/+}$ and RANK$^{-/-}$ tumor transplants, but a moderate increase in IFNγ production in the lymph node T cells was observed in the RANK$^{-/-}$ tumor transplants (Supplementary Fig. 3b).

Next, we investigated the effects on the tumor immune infiltrates after systemic pharmacological inhibition of RL (RANK-Fc treatment 10 mg/kg three times per week, for 4 weeks) in serial tumor transplants from PyMT mice (Supplementary Fig. 3c)[10]. No significant changes in the total number of TILs upon RL inhibition were observed (Supplementary Fig. 3d, e). However, after RL inhibition, the frequency of infiltrating CD8$^+$ T cells increased (Supplementary Fig. 3d) and CD4$^+$ T cells decreased (Supplementary Fig. 3e), leading to a lower CD4$^+$/CD8$^+$ ratio (Supplementary Fig. 3d, e). An increased infiltration by CD8$^+$ T cells in tumors continuously treated with RL inhibitor was also observed by IHC (Fig. 2e, f). Together, these evidences demonstrate that genetic or pharmacologic inhibition of RANK signaling increases CD8$^+$ T-cell tumor infiltration.

CD8$^+$ T and NK cells have been shown to drive tumor cell cytotoxicity[20]; therefore, depletion experiments were performed in RANK$^{+/+}$ and RANK$^{-/-}$ tumor transplants to confirm their involvement (Fig. 2g). Depletion of CD8$^+$ T cells, but not of NK cells, rescued the delayed tumor formation observed in RANK$^{-/-}$ transplants with minor effects on RANK$^{+/+}$ transplants (Fig. 2h). CD8$^+$ T- and NK-cell depletions were corroborated in blood samples and tumor infiltrates (Supplementary Fig. 4a, b). CD8$^+$ T-cell depletion resulted in increased NK-cell frequency in tumors and, conversely, NK-cell depletion led to increased CD8$^+$ T-cell

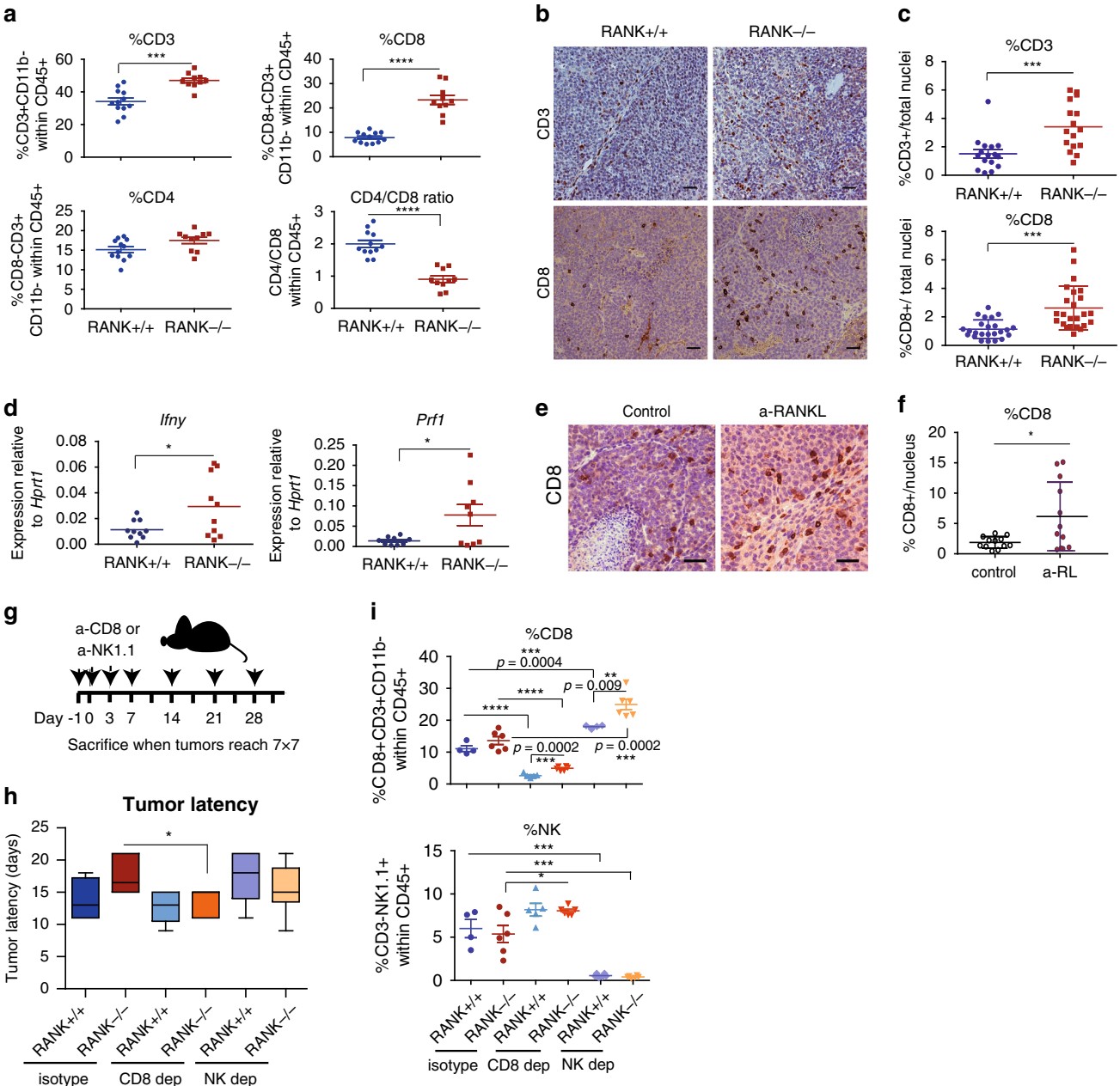

**Fig. 2 RANK loss in tumor cells leads to increased CD8$^+$ Tcell tumor infiltration that mediates the delayed tumor latency of RANK$^{-/-}$ tumors. a** Graphs showing the percentage of T cells (CD3$^+$CD11b$^-$ within CD45$^+$; ***$p = 0.0001$), CD8 (CD8$^+$CD3$^+$CD11b$^-$ within CD45$^+$; ****$p < 0.0001$), CD4 (CD8-CD3$^+$CD11b$^-$ within CD45$^+$; $p = 0.0503$), and the CD4/CD8 ratio (****$p < 0.0001$) in RANK$^{+/+}$ ($n = 12$) or RANK$^{-/-}$ ($n = 10$) tumor cells injected in syngeneic C57BL/6 mice[#]. Representative images (**b**) and quantification (**c**) of CD3$^+$ ($n = 4$ tumors, ***$p = 0.0009$) and CD8$^+$ cells ($n = 6$ tumors, ***$p = 0.0001$) in RANK$^{+/+}$ and RANK$^{-/-}$ tumor transplants as assessed by IHC. Scale = 25 μm. Tumors derived from three independent primary tumors were used. Each dot represents one picture[#]. **d** Prf1 and Ifnγ mRNA levels relative to Hprt1 of whole tumors from RANK$^{+/+}$ and RANK$^{-/-}$ transplants in syngeneic C57BL/6 mice ($n = 10$; Prf1 *$p = 0.0286$, Ifnγ *$p = 0.0360$)[#]. **e, f** Representative images (**e**) and quantification (**f**) of CD8$^+$ cells in RANK$^{+/+}$ control and anti-RANKL-treated tumors from second transplants as assessed by IHC. Scale = 25 μm. Each dot represents one picture ($n = 12$ pictures, $n = 3$ tumors, *$p = 0.0168$)[#]. **g** Schematic overview of CD8 (300 μg, clone 53-5.8) and NK1.1 (200 μg, clone PK136) treatments in orthotopic RANK$^{+/+}$ and RANK$^{-/-}$ tumor transplants. Animals were treated i.p. on days −1, 0, 3, and 7 after tumor cell injection and then once per week until the day of killing, when tumors were >0.5 cm$^2$. **h** Latency to tumor onset of RANK$^{+/+}$ and RANK$^{-/-}$ tumor cells implanted in syngeneic C57BL/6 animals and treated with anti-CD8 or anti-NK1.1 depletion antibodies ($n = 6$) or corresponding isotype control ($n = 4$ for RANK$^{+/+}$ and $n = 6$ for RANK$^{-/-}$). Box and whisker plots (box represents the median and the 25th and 75th percentiles, whiskers show the largest and smallest values) and significant $t$-test two-tailed $p$-values are shown (*$p = 0.05$). **i** Graphs showing the percentage of infiltrating CD8 T cells (CD8$^+$CD3$^+$CD11b$^-$ within CD45$^+$) and NK (NK1.1$^+$CD3$^-$ within CD45$^+$). Each dot represents one tumor ($n = 4$ control and NK-depleted RANK$^{+/+}$ tumors; $n = 5$ CD8-depleted RANK$^{+/+}$ tumors; and $n = 6$ RANK$^{-/-}$ control, NK- and CD8-depleted tumors)[#]. [#]Mean, SEM and $t$-test two-tailed $p$-values are shown (*$p < 0.05$; **$0.001 < p < 0.01$; ***$0.001 < p < 0.0001$; ****$p < 0.0001$). For **a** and **d**, each dot represents one tumor analyzed at the endpoint (>0.2 cm$^2$). Data for tumor transplants derived from two representative primary tumors in two independent experiments.

infiltration (Fig. 2i). These results suggest that CD8[+] T cells mediate the anti-tumorigenic response induced by RANK loss in tumor cells, and that the exacerbated T-cell response in RANK[−/−] tumors is responsible for the delay in tumor formation.

**RANK[+] tumor cells promote immunosuppression through neutrophils.** To clarify the intercellular crosstalk involved in the observed phenotypes we cultured three-dimensional (3D) tumor acini from RANK[+/+] and RANK[−/−] tumor transplants for 72 h, and measured the levels of cytokines and chemokines in the culture supernatants (Supplementary Data 5). Fewer cytokines/chemokines were more abundant in RANK[−/−] than in RANK[+/+] tumor supernatants and included the following: (i) eotaxin 1, which is involved in eosinophil recruitment; (ii) CD40, which enhances T-cell responses; and (iii) B lymphocyte chemoattractant (BLC), which controls B-cell trafficking[28] (Fig. 3a). However, no significant differences in the frequencies of eosinophils or B cells were found in RANK[−/−] as compared to RANK[+/+] tumor transplants (Supplementary Fig. 3a). In supernatants derived from RANK[+/+] tumor acini, many cytokines were upregulated including stromal cell-derived factor-1α, macrophage inflammatory protein-1α, interleukin (IL)-1α, stem cell factor, tumor necrosis factor-α, IL-13, macrophage colony-stimulating factor, IL-10, IL-4, IL-17, and IL-1β (Supplementary Data 5 and Fig. 3a). These various cytokines/chemokines are characteristic of an immunosuppressive microenvironment and have a wide-range of actions, including myeloid cell recruitment[28]. The mRNA expression levels of *Il-1β* and *Caspase-4*, which mediates the activation of pre-IL1-β in the inflammasome[29], were also higher in RANK[+/+] tumors, whereas *s100a9*, a gene related to neutrophil stimulation and migration, showed a tendency to increase[30] (Fig. 3b). These changes may contribute to the increased infiltration of TANs observed in RANK[+/+] tumors (Fig. 1f, g and Supplementary Figs. 1c and 2d) and the suppression of T-cell immunity as previously reported[31,32]. In fact, the percentage of TANs (Ly6G[+]) and that of CD8[+] T cells were negatively correlated in the mouse tumors (Fig. 3c).

To confirm the crosstalk between RANK activation in BC cells and neutrophils, we adopted an independent experimental approach by modulating RANK expression levels in human BC cells and directly testing in co-culture assays whether this influenced neutrophil survival and activation. MCF7 luminal BC cells that had undetectable RANK expression and were unresponsive to RL stimulation, were infected with RANK-overexpressing vectors (Supplementary Fig. 4c). Conversely, HCC1954 basal-like HER2[+] cells, which, despite the low levels of RANK expression, are responsive to RL stimulation, were infected with two different short hairpin RNAs to downregulate RANK (Supplementary Fig. 4c). Corresponding changes in RANK expression and downstream targets (*BIRC3*, *ICAM1*, *NFKB2*, and *RELB*) in these BC cells were confirmed by RT-PCR (Supplementary Fig. 4c).

BC cells were stimulated with RL for 1 h before co-culturing with neutrophils isolated from blood of healthy human donors (Supplementary Fig. 4d). MCF7-RANK tumor cells and HCC1954-shSCR cells increased neutrophil survival more than did their corresponding tumor cells lacking RANK (MCF7-GFP and HCC1954 shRANK, respectively) (Supplementary Fig. 4e). Conditioned medium (CM) from BC cells with higher level of RANK expression and activation was enough to increase the survival of neutrophils significantly more than CM from cells with low RANK (Fig. 3d). These neutrophils also presented a more mature/active phenotype based on the increased CD11b levels (Fig. 3e)[33].

Finally, to confirm whether neutrophils are involved in the observed differences in latency between RANK[+/+] and RANK[−/−] tumor transplants and the crosstalk with T cells, Ly6G depletion

assays were performed (Fig. 3f). Neutrophil depletion significantly delayed tumor appearance in RANK[+/+] transplants with no effects in RANK[−/−] transplants (Fig. 3g). Neutrophil depletion was confirmed in blood samples (Supplementary Fig. 4f, g). The frequency of TANs after depletion was reduced in RANK[+/+] but not in RANK[−/−] tumor transplants, in which TAN infiltration was much lower (Fig. 3h). Neutrophil depletion led to a significant increase in TILs, CD4[+], and CD8[+] T cells, and to a decrease in the frequency of TAMs infiltrating RANK[+/+] transplants to levels comparable with those found in RANK[−/−] transplants (Fig. 3h). A trend to increased levels of total leukocyte infiltration was also observed after neutrophil depletion ($p = 0.06$, Fig. 3h).

Altogether, these results suggest that RANK activation in tumor cells induces an immunosuppressive microenvironment that favors neutrophil survival, thus restricting T-cell immunity.

**RL inhibition in tumor cells increases responsiveness to immunotherapy.** Despite the stronger anti-tumor immune response, RANK[−/−] tumors eventually evade the immune response and grow. Increased expression of checkpoint regulators such as PD-1 in lymphoid cells and CTLA4 in CD4[+] T cells was found in RANK[−/−] relative to RANK[+/+] tumors (Fig. 4a). The level of PD-L1 expression in RANK[−/−] tumor cells was also higher than in RANK[+/+] tumors (Fig. 4a). Tregs (FoxP3[+] CD25[+] CD4[+] CD11b[−]) were more frequent in RANK[−/−] than in RANK[+/+] tumors, possibly as a result of the enhanced cytotoxic response, as reported elsewhere[34] (Fig. 4a). These results suggest that the exacerbated T-cell response in RANK[−/−] tumors may facilitate the induction of negative immune-checkpoint regulators and Tregs, evading immune surveillance and allowing tumor growth. This prompted us to investigate the effects of anti-PD-L1 and/or anti-CTLA4 checkpoints inhibitors in combination with the loss of RANK signaling. In RANK[+/+] tumors early treatment (72 h after tumor implantation) with anti-RL did not affect tumor growth; however, anti-CTLA4 combined with anti-RL reduced tumor growth to a greater extent than did single anti-CTLA4 treatment (28.5% of implanted tumors did not even grow) (Fig. 4b, c). No benefit of combining anti-RL and anti-PD-L1 compared to anti-PD-L1 alone was observed in RANK[+/+] tumors in the early setting (Fig. 4b, c).

Early treatment with anti-CTLA4, but not with anti-PD-L1 or anti-RL, significantly attenuated RANK[−/−] tumor growth (66.7% of implanted tumors did not grow) compared with the isotype-treated control (Fig. 4d). Addition of anti-RL did not improve the response to anti-CTLA4 (or anti-PD-L1) in RANK[−/−] tumors as did in RANK[+/+] tumors, suggesting that the augmented benefit of the anti-RL/anti-CTLA4 combination was driven by inhibition of RANK signaling in tumor cells (Fig. 4d).

Next, we tested the effect of checkpoint inhibitors on the growth of already palpable, actively growing tumors (Fig. 4e). None of the RANK[+/+] tumors responded to anti-PD-L1 or anti-RL as single agents but their combination significantly reduced tumor growth in 50% of the tumors (Fig. 4f). Anti-RL did not improve the response to anti-CTLA4 (Fig. 4f). In tumors lacking RANK, anti-PD-L1 treatment was more efficient than anti-CTLA4, but no improvement was observed after the addition of anti-RL (Fig. 4g), in contrast with the observations on RANK[+/+] tumors.

Collectively, these results demonstrate that in this luminal-like BC, RL inhibition improves the anti-tumor response to anti-CTLA4 (in the early setting) and anti-PD-L1 (for established tumors) through inhibition of RANK signaling in the tumor cells.

**A short course of denosumab treatment in early-stage BC increased TILs.** To confirm the immunomodulatory role of RANK pathway inhibition in the clinical setting, we analyzed

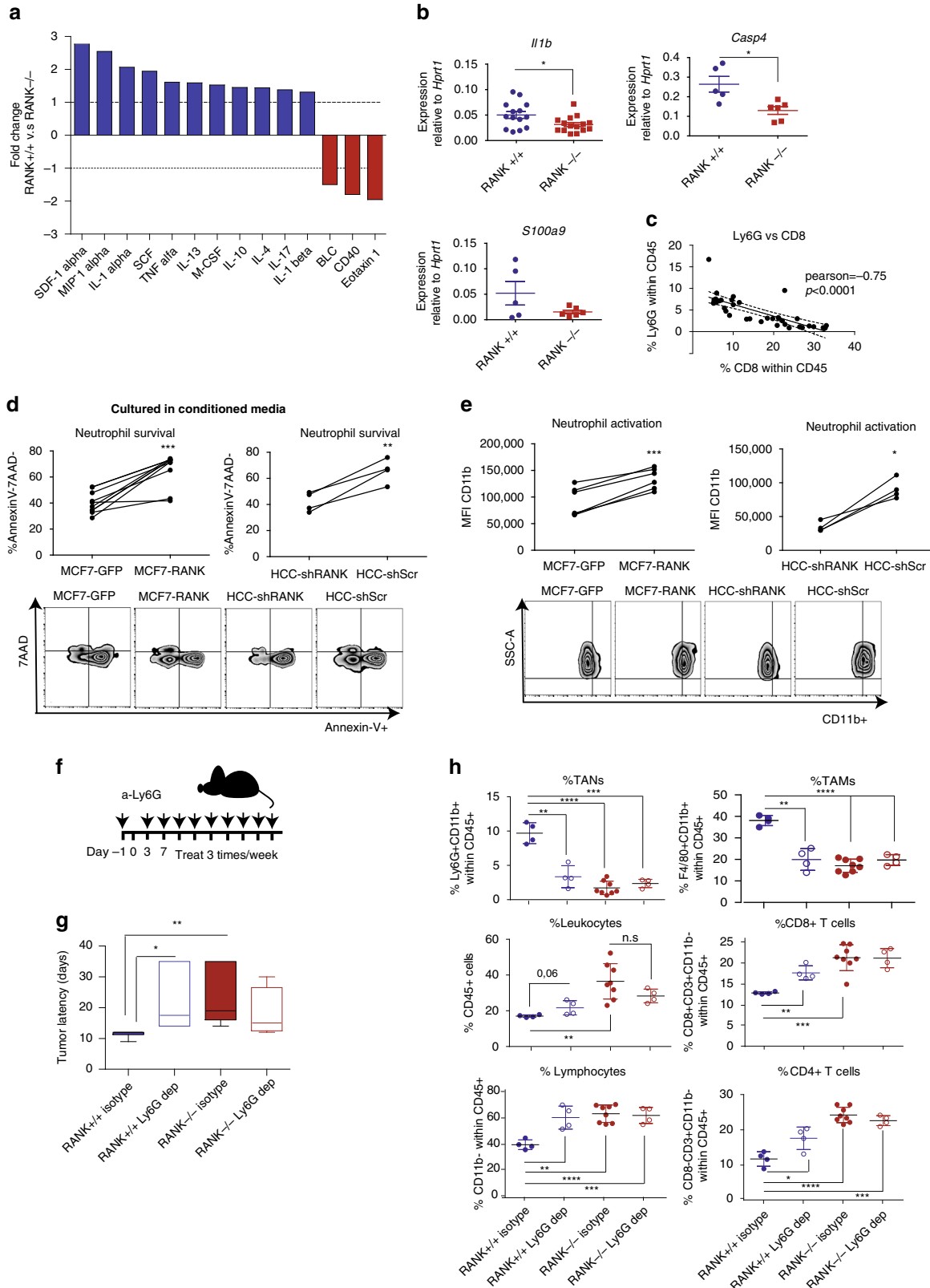

denosumab-treated BC clinical samples from the D-BEYOND study (NCT01864798): a prospective, pre-operative window-of-opportunity, single-arm, multi-center trial assessing the effect of denosumab in premenopausal women with early-stage BC. Twenty-seven patients were included in this study and received two doses of denosumab 120 mg subcutaneously 1 week apart,

followed by surgery. The median time interval between the first administration of denosumab and surgery was 13 days. No serious adverse events (AEs) were reported. All non-serious AEs are summarized in Supplementary Data 6, the most frequent being arthralgia (4/27 patients, 14.8%). Table 1 summarizes the clinicopathological features of the 24 patients subsequently analyzed.

**Fig. 3 Neutrophils recruited by the proinflammatory cytokine/chemokine milieu driven by RANK restrict T-cell immunity. a** Cytokines/chemokines in the supernatant of RANK$^{+/+}$ and RANK$^{-/-}$ tumor 3D acini cultured during 72 h, expressed as the magnitude of change between RANK$^{+/+}$ and RANK$^{-/-}$ tumor acini (pool of 3 tumors, $n = 1$). See also Supplementary Data 5. **b** *Il1b, Casp4*, and *S100a9* mRNA levels relative to *Hprt1* of whole tumors from RANK$^{+/+}$ and RANK$^{-/-}$ transplants in syngeneic C57BL/6 mice ($n = 14$ for *Il1b*, $*p = 0.005$; $n = 5$ RANK$^{+/+}$ tumors, $n = 6$ RANK$^{-/-}$ tumors for *Casp4*, $p = 0.011$; and *S100a9*, $p = 0.12$). Two representative primary tumors of two independent experiments were used[#]. **c** Correlation between the frequency of TANs (Ly6G$^+$ Ly6C$^+$ CD11b$^+$) and CD8$^+$-T cells (CD8$^+$ CD3$^+$ CD11b$^-$) infiltrates in tumor transplants. Pearson's correlation coefficients ($r$) associated probabilities are shown ($p < 0.0001$). **d** Percentage of Annexin V–7AAD$^-$ neutrophils ($n = 5$, 2 healthy donors) cultured with conditioned media (CM) from the indicated RL-treated tumor cells. CM was added (1 : 1) to human neutrophil cultures for 24 h. Paired *t*-test with one-tailed *p*-value is shown (***$p = 0.0002$, **$p = 0.009$). **e** Mean fluorescence intensity (MFI) of CD11b$^+$ neutrophils ($n = 4$, 2 healthy donors) cultured in CM from the indicated RL-treated tumor cells. CM was added (1 : 1) to human neutrophils cultures for 24 h. Paired *t*-test with one-tailed *p*-value is shown (***$p = 0.0004$, *$p = 0.01$). **f** Schematic overview of TAN (Ly6G$^+$) depletion experiments in orthotopic RANK$^{+/+}$ and RANK$^{-/-}$ tumor transplants. Anti-Ly6G (clone 1A8) was administered i.p. before tumor cell injection (400 μg) and then once per week (100 μg) until the day of killing. **g** Latency to tumor formation of RANK$^{+/+}$ and RANK$^{-/-}$ tumor cells orthotopically implanted in syngeneic C57BL/6 animals and treated with anti-Ly6G depletion antibody or isotype control ($n = 4$ control and neutrophil-depleted RANK$^{+/+}$ tumors, $n = 8$ control RANK$^{-/-}$ tumors, $n = 4$ neutrophil-depleted RANK$^{-/-}$ tumors). Box and whisker plots (box represents the median and the 25th and 75th percentiles, whiskers show the largest and smallest values) and *t*-test two-tailed *p*-values are shown. (*$p = 0.028$; **$p = 0.007$). **h** Graphs showing the percentage of TANs (Ly6G$^+$ CD11b$^+$, **$p = 0.0012$; ***$p = 0.0003$; ****$p < 0.0001$), leukocytes (CD45$^+$; **$p = 0.034$), lymphocytes (CD11b$^-$; **$p = 0.048$; ***$p = 0.0008$; ****$p < 0.0001$), TAMs (F4/80$^+$ CD11b$^+$, **$p = 0.0019$; ****$p < 0.0001$), CD8$^+$ T cells (CD8$^+$ CD3$^+$ CD11b$^-$, ***$p = 0.0003$, **$p = 0.0014$), and CD4$^+$ T cells (CD8$^-$ CD3$^+$ CD11b$^-$, *$p = 0.0213$; ***$p = 0.001$; ****$p < 0.0001$) ($n = 4$ control and neutrophil-depleted RANK$^{+/+}$ tumors, $n = 8$ control RANK$^{-/-}$ tumors, $n = 4$ neutrophil-depleted RANK$^{-/-}$ tumors)[#]. [#]Each dot represents one tumor. Mean, SEM, and *t*-test two-tailed *p*-values are shown (*$p < 0.05$; **$p < 0.01$; ***$p < 0.001$; ****$p < 00001$). Tumors of similar size were analyzed at endpoint (>0.2 cm$^2$). For **d**, **e**, each dot represents a technical replicate from healthy donors. Representative dot blots are shown below.

In brief, the median age at diagnosis was 45 years (range, 35–51 years); tumors of 19 patients were hormone receptor positive (79.2%), 4 were HER2$^+$ (16.7%), and 1 was triple negative (4.2%). After treatment, serum levels of soluble homotrimeric form of RL (sRL) (unbound to denosumab) and C-terminal telopeptide (CTX), a surrogate marker for denosumab activity, decreased in all patients evaluated ($P < 0.001$, Fig. 5a), confirming the target inhibition. Given its correlation with clinical response in luminal BC[35–37], the primary study endpoint was a geometric mean (GM) decrease in the percentage of Ki-67-positive cells. Secondary endpoints included tumor cell survival assessed by cleaved caspase-3, as well as tumor immune infiltration. There was no significant reduction in the percentage of Ki-67-positive cells (GM change from baseline; 1.07, 95% confidence interval (95% CI) 0.87–1.33, $P = 0.485$, Fig. 5a) and no absolute Ki-67 or cleaved caspase-3 responders were identified (Fig. 5a and Supplementary Fig. 5a).

Collectively, these data confirm that a short course of denosumab was associated with effective systemic RL inhibition, but not with a reduction in tumor proliferation or survival.

Next, we assessed the effect of denosumab on tumor immune infiltration in 24 available paired samples. Of note, similar to our preclinical model, we observed a significant increase in stromal and intratumoral lymphocyte levels after short exposure to denosumab (GM change from baseline: 1.75, 95% CI 1.28–2.39, $P = 0.006$ and 1.59, 95% CI 1.14–2.21, $P = 0.008$, respectively, Fig. 5b, c and Supplementary Fig. 5a). In particular, 11/24 patients (45.8%), including 6/14 luminal A, 3/5 luminal B, and 2/4 HER2$^+$ cases, showed an immunomodulatory response defined as a ≥10 percent increase in stromal TILs (sTILs) in tumor samples, and therefore they were considered responders. Analyses of the percentage of Ki-67$^+$ TILs suggested a trend to increase after denosumab treatment, particularly in responders (7/11) (Fig. 5b).

The composition of the immune infiltrate associated with denosumab treatment was analyzed by IHC in 23 available pairs of pre- and post-denosumab treatment tumor tissues (Fig. 5b and Supplementary Fig. 5a, b). These analyses revealed a significant increase in the percentage of T (CD3$^+$) and B (CD20$^+$) cells after denosumab treatment (GM change from baseline: 1.68, 95% CI 1.18–2.40, $P = 0.006$ and 1.62, 95% CI 1.09–2.40, $P = 0.019$, respectively) and increased levels of CD8$^+$ T cells, validating our

preclinical observations (GM change from baseline: 1.59, 95% CI 1.14–2.21, $P = 0.008$). Moreover, there was a significant decrease in FOXP3$^+$/CD4$^+$ Tregs cell frequency (GM change from baseline: 0.63, 95% CI 0.49–0.83, $P = 0.002$, Fig. 5b), even in patients with no increase in TILs. No significant differences in macrophage infiltration (CD68$^+$ or CD163$^+$) were observed (Fig. 5b and Supplementary Fig. 5a). Intratumoral immune population abundance was also quantified, and an increase of TILs and CD3$^+$ T cells was observed (Supplementary Fig. 5a). These findings were illustrated using multiplex IHC of the top four tumors associated with the highest TIL increase (Fig. 5c).

To investigate the biological effect of denosumab in early BC further, we performed RNA sequencing (RNA-seq) on 22 available pre- and posttreatment tumor samples and identified 379 genes that were differentially expressed (Supplementary Data 7). In addition, we performed RNA-seq on 11 available pre- and post-treatment normal mammary samples. Only ten genes were differentially expressed between pre- and posttreatment normal samples (Supplementary Data 8) and all of them were also differentially expressed in the tumor tissue (Supplementary Data 7). Pathway analysis using GO and GAGE in the tumor-derived RNA-seq data revealed the enrichment of several genes related to immune activation, immune cell migration, and cytokine-mediated signaling pathways (Fig. 5d and Supplementary Data 9 and 10). In line with these findings, the expression levels of several chemokines were increased after treatment, including that of the well-known CD8$^+$ T-cell chemoattractants CCL4 and CXCL10[38,39] (Supplementary Fig. 5c). No significant changes in RANK/RL at the protein (IHC) (Supplementary Fig. 5d, e) or at the gene expression levels (RNA-seq) (Supplementary Data 7 and 8) were found. Of note, no differences in genes related to immature mammary epithelial cell (MEC) populations (*ALDH1*) or related to estrogen receptor (ER) pathway (*ESR1, PR, BCL2*) both in tumor and normal samples, were observed (D-BEYOND secondary endpoints) (Supplementary Data 7 and 8).

To further explore the impact of denosumab treatment on the immune cell landscape of BC we used CIBERSORT[40], a deconvolution method for inferring immune cell content from gene expression data. Consistent with the IHC results, this analysis confirmed the increase in the relative frequencies of CD8$^+$ T cells, B cells, and CD4$^+$ T cells, and the decrease in the

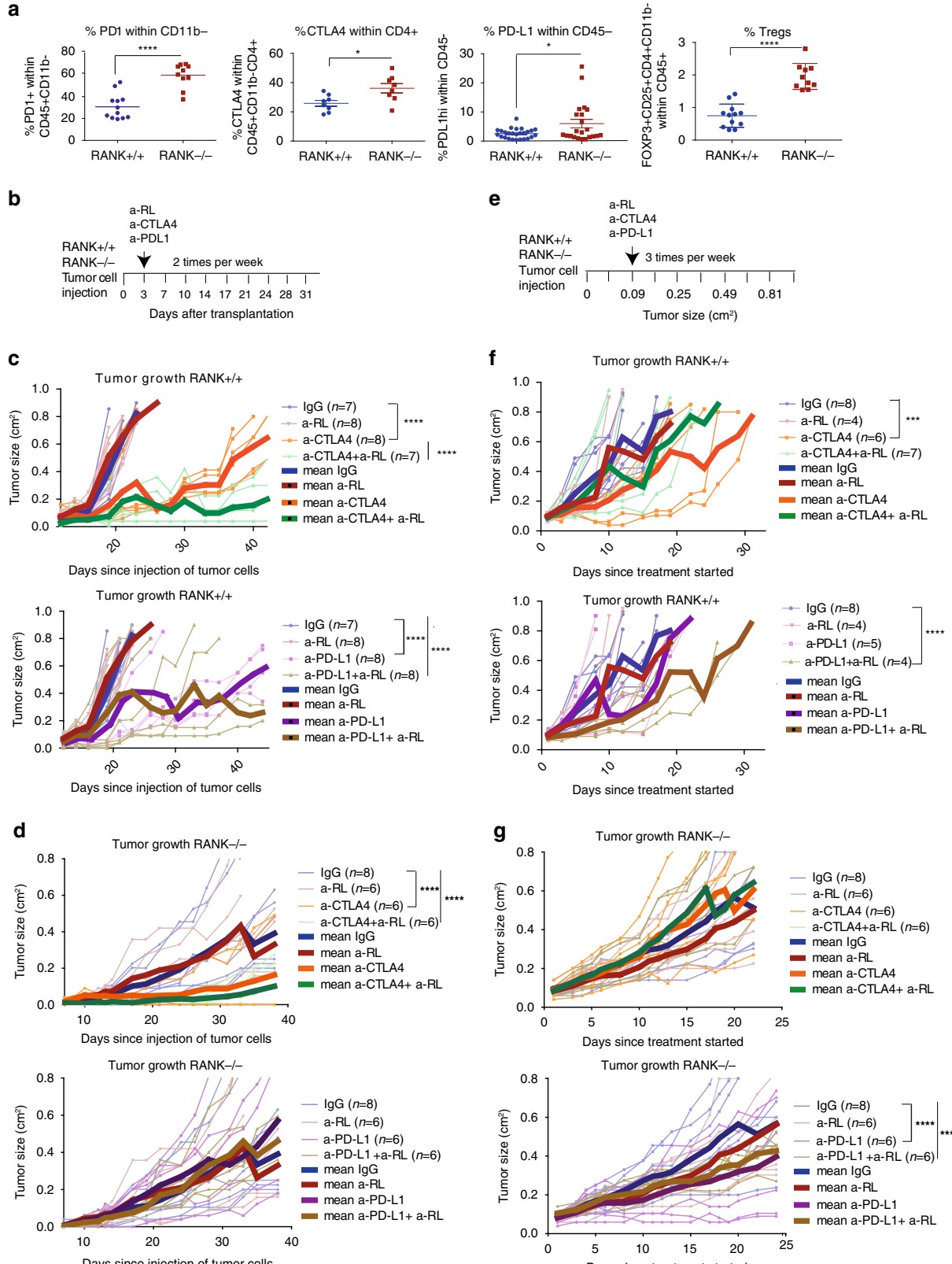

frequencies of Tregs after denosumab treatment (Supplementary Fig. 5f). Despite the overall increase in immune infiltration, the relative frequency of macrophage infiltration was reduced after denosumab, particularly in responders (8/11) (Supplementary Fig. 5f), as observed in the mouse models. No significant changes in NK cells, dendritic cells, mast cells, neutrophils, and eosinophils were noted, because these populations may be too scarce to be captured properly by this method (Supplementary Fig. 5f). Of note, after denosumab treatment, neutrophils correlated negatively with sTILs (Supplementary Fig. 5g), and

**Fig. 4 RANKL pharmacological inhibition reinforces anti-CTLA4 and anti-PD-L1 anti-tumor response in RANK$^{+/+}$ but not in RANK$^{-/-}$ tumors. a** Graphs showing the percentage of PD-1$^+$ cells within CD11b$^-$ lymphocytes ($n = 12$ RANK$^{+/+}$ tumors, $n = 10$ RANK$^{-/-}$ tumors; PD-1$^+$ within CD11b$^-$ CD45$^+$; ****$p < 0.0001$), CTLA4 within CD4$^+$ T cells ($n = 8$; CTLA4 within CD3$^+$ CD8$^-$CD11b$^-$CD45$^+$; *$p = 0.0166$), PD-L1 within tumor CD45$^-$ cells ($n = 26$ RANK$^{+/+}$ tumors, $n = 22$ RANK$^{-/-}$ tumors; *$p = 0.017$), and Tregs ($n = 12$ RANK$^{+/+}$ tumors, $n = 10$ RANK$^{-/-}$ tumors; FoxP3$^+$ CD25$^+$ CD4$^+$ CD11b$^-$ within CD45$^+$; ****$p < 0.0001$) in RANK$^{+/+}$ and RANK$^{-/-}$ transplants in syngeneic C57BL/6 mice. Each dot represents an individual tumor transplant derived from two to five different primary tumors. Mean, SEM, and $t$-test two-tailed $p$-values are shown (*$p < 0.05$; ****$p < 0.0001$). **b** Experimental scheme for early treatments with anti-RANKL (a-RL), anti-CTLA4, anti-PD-L1, or their respective isotype controls (rat IgG2A and mouse IgG2b). All treatments were administered i.p, two times/week, and started 3 days after injection of RANK$^{+/+}$ and RANK$^{-/-}$ tumor cells into the mammary gland of syngeneic C57BL/6 mice. **c, d** Tumor growth curves for early treatments (scheduled as in Fig. 4b) of RANK$^{+/+}$ (**c**) and RANK$^{-/-}$ (**d**) tumor cells injected in syngeneic C57BL/6. Each thin curve represents one single tumor. Each thick curve represents the mean of all the tumors that received the specific treatment. Linear regression analysis was performed and a two-tailed $p$-value was calculated to compare the tumor growth slopes after the specified treatments (****$p < 0.0001$). **e** Experimental scheme for late treatments with anti-RL, anti-CTLA4, anti-PD-L1, or their respective isotype controls (rat IgG2A and mouse IgG2b). All treatments were administered i.p., three times/week, and started when transplanted tumors reached a size of 0.09 cm$^2$. **f, g** Tumor growth curves for late treatments (scheduled as in Fig. 4e) of RANK$^{+/+}$ (**f**) and RANK$^{-/-}$ (**g**) tumor cells injected in syngeneic C57BL/6. Each thin curve represents one single tumor. Each thick curve represents the mean of all the tumors that received the specific treatment. Linear regression analysis was performed and a two-tailed $p$-value was calculated to compare the tumor growth slopes after the specified treatments ***$p = 0.0002$; ****$p < 0.0001$).

**Table 1 Clinicopathological features of the 24 evaluable patients.**

| N | | 24 |
|---|---|---|
| Interval surgery-Denosumab | Median days (range) | 13 (9–21) |
| Age | Median years (range) | 44 (35–51) |
| Size | >2 cm | 11 (45.8%) |
| Nodal status | Positive | 4 (16.7%) |
| Histological grade | High | 8 (33.3%) |
| Molecular subtypes | LumA | 10 (41.7%) |
| | LumB | 9 (37.5%) |
| | HER2 | 4 (16.7%) |
| | TNBC | 1 (4.2%) |
| Immune response | Percentage of patients | 11 (45.8%) |

the neutrophil chemotaxis and migration pathways were modulated after denosumab treatment (Supplementary Data 9), supporting the preclinical findings.

To ensure that these changes are specific to denosumab treatment and not a consequence of the presurgical biopsy procedure, we interrogated the publicly available gene expression data of patients from the control arm (untreated) of the Peri Operative Endocrine Therapy - Individualizing Care (POETIC) study, a large BC window-of-opportunity study evaluating the role of perioperative aromatase inhibitor, for which gene expression data were obtained from presurgical biopsies and surgical specimens. Similar to the D-BEYOND study, biopsies were taken at diagnosis and 2 weeks later, at the time of surgery. The comparison of surgery and biopsy samples from the POETIC study did not reveal any enrichment of immune cells assessed by CIBERSORT or an immune pathway, as assessed by GAGE analyses (Supplementary Fig. 5h and Supplementary Data 11). Together, our results indicate that a short course of denosumab enhances immune infiltration as determined by the increased levels of TILs, B and T lymphocytes, and CD4$^+$ and CD8$^+$ T cells in luminal and HER2$^+$ breast tumors, validating the clinical relevance of the findings in the preclinical models.

**RANK pathway activation in tumors and circulating sRL levels predict denosumab's immune effect.** Finally, we investigated the baseline features associated with the immunomodulatory effect of denosumab. We identified 11 responder (R) cases, defined by a ≥10% increase in TIL infiltration after denosumab treatment and 13 non-responder (NR) cases. No associations were found between any baseline clinicopathological features and the

immune modulation induced by denosumab (Supplementary Data 12). Of the characteristics compared between R and NR patients, high sRL serum levels, a high percentage of Tregs measured by CIBERSORT, and the presence of intratumoral FOXP3+ cells measured by IHC, were significantly associated with increased TIL infiltration after denosumab treatment (Fig. 5e and Supplementary Data 12). CD20 IHC staining at baseline was also associated with response, but this finding was not corroborated by CIBERSORT (Supplementary Data 12). A differential gene expression analysis using RNA-seq data from biopsy samples evidenced 42 genes expressed at higher levels in R than in NR, including *FOXP3*, *IL7R*, *MS4A1* (CD20), *CD28*, and *IFNG* (Fig. 5f and Supplementary Data 13), and the enrichment of genes involved in lymphocyte activation and immunoglobulin production in R patients (Supplementary Data 14), which may be indicative of an enhanced immune response.

RANK and RL expression determined by IHC was not predictive of the immunomodulatory effects of denosumab (Supplementary Fig. 6a). However, as it has been reported that RANK IHC is an unreliable tool to detect RANK protein on breast tumor samples[41], we computed RANK and RL metagenes to increase the potency and reliability of RANK and RL detection. These metagenes included the expression levels of the top 100 genes that are co-expressed at baseline with *RANK* and *RANKL* mRNA, respectively (see "Methods" and Supplementary Data 15). Importantly, high expression level of RANK metagene in the tumors at baseline (Fig. 5g), but neither RL metagene nor individual gene expression of *RANK* or *RANKL*, is predictive of denosumab-induced immune response (Supplementary Fig. 6b).

GO analyses showed that the RANK metagene includes genes associated with nuclear factor-κB (NF-κB) pathway activation, as well as with immune response (Supplementary Fig. 6c). Indeed, the RANK metagene strongly correlated with several public signatures of the RANK and NF-κB pathways, as well as with RL-induced genes in mouse MECs (wild type (WT) and Rank overexpressing) and PyMT tumor cells (Supplementary Fig. 6d and Supplementary Data 16). These results demonstrate that RANK metagene captures RANK pathway activation and support the relevance of the PyMT model. Accordingly, tumors responding to denosumab presented at baseline higher scores for these RL-driven genes in mouse MECs and PyMT tumor cells (Fig. 5g and Supplementary Fig. 6e), and RANK and NF-κB pathway gene signatures (Supplementary Fig. 6f). Thus, tumors with increased RANK pathway activation at baseline are more likely to show increased TILs after RL inhibition, corroborating the preclinical findings: inhibition of RANK signaling in tumor cells contributes to the immunomodulatory effect of denosumab in BC.

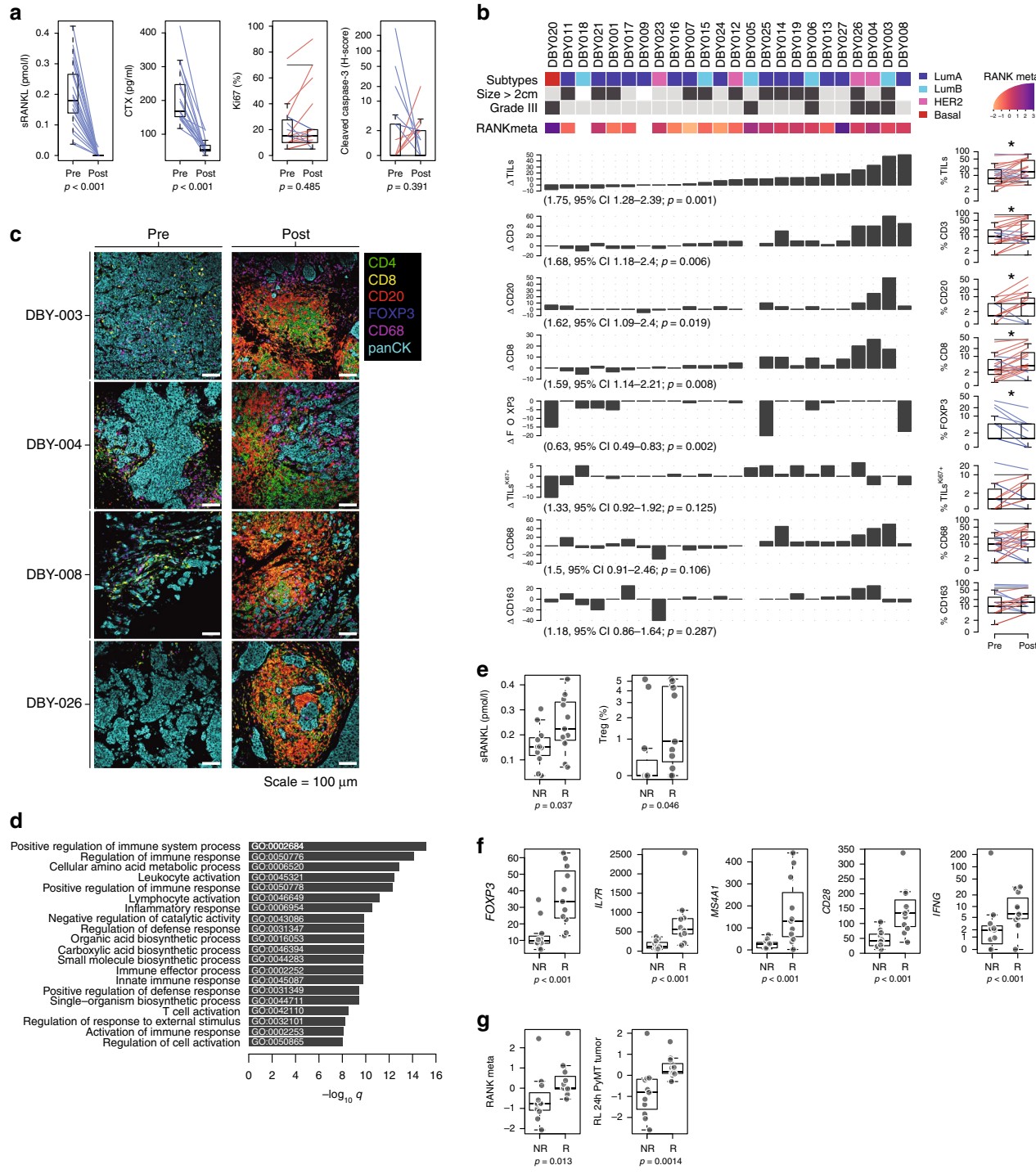

Together, these results indicate that higher RANK pathway activation, soluble RL, and the presence of Tregs at baseline are predictive biomarkers of the immunomodulatory response induced by denosumab in BC patients.

## Discussion

Several studies have shown the prognostic and predictive value of TILs, especially in HER2[+] and triple-negative BC[42,43]. However, TILs continue to be infrequent in most luminal breast tumors. The identification of a therapy that could convert immune "cold" tumors into "hot" ones would represent a major step towards the development of immune-related therapies. Based on our clinical and preclinical findings, denosumab appears to be just this type of promising therapeutic agent. This question is particularly relevant for luminal BC, which is poorly infiltrated and insensitive to immunotherapies.

The results of the D-BEYOND clinical trial provide strong evidence of the immunomodulatory effect of denosumab in luminal early BC and identify predictive biomarkers of response. The mouse genetic studies demonstrate that inhibition of RANK signaling in the tumor cells increases TILs and CD8[+] T-cell infiltration, and attenuates tumor growth. Mechanistically we found that activation of RANK signaling in tumor cells induces a proinflammatory microenvironment that favors survival of TANs and restricts T-cell anti-tumor response.

**Fig. 5 The immunomodulatory role of anti-RANKL in BC. a** Change from baseline in serum levels of free-sRANKL ($n = 23$, $p = 2.384\text{e-}07$) and CTX ($n = 17$, $p = 1.526\text{e-}05$) (significance assessed by the two-tailed sign test), the percentage of Ki-67-positive cells ($p = 0.485$) and the staining of activated caspase-3 ($p = 0.391$) (H-score) ($n = 24$) (significance assessed by two-tailed paired $t$-tests). Boxplots display median line, IQR boxes, 1.5 × IQR whiskers, and data points. **b** Each bar plot shows the change from baseline (Δ; post- minus pretreatment values) of the immune parameters assessed using HE (TILs) and IHC (CD3, CD20, CD8, FOXP3, proliferative TILs (TILs$^{Ki67+}$), CD68, and CD163). Each bar represents one patient, which are ranked by their increase in stromal TIL levels. Geometric mean changes, 95% CIs, and $p$-values are shown below each bar plot. For each measured parameter, the corresponding boxplot is displayed on the right-hand side. Boxplots display median line, IQR boxes, 1.5 × IQR whiskers, and data points. Tumor characteristics and tumor RANK metagene expression at baseline are shown above. p; $p$-values derived from two-tailed paired $t$-tests (*$p < 0.05$)#. **c** Representative micrographs of multiplex IHC of pre- and posttreatment tumor sections from the four patients with the highest immunomodulatory response. White scale bar, 100 μm. **d** Top 20 significantly enriched pathways after denosumab treatment, identified by GAGE. **e** Comparison of baseline serum levels of sRANKL between non-responders (NR; $n = 13$) vs. responders (R; $n = 11$) and comparison of baseline percentage of regulatory T cells (Tregs) as inferred from CIBERSORT. Boxplots display median line, IQR boxes, 1.5 × IQR whiskers, and data points. Significance determined by the two tailed Mann–Whitney U-test. **f** Comparison of baseline mRNA expression levels of indicated genes (normalized counts) between non-responder (NR; $n = 11$) and responder (R; $n = 11$) groups. Boxplots display median line, IQR boxes, 1.5 × IQR whiskers, and data points. Significance determined by the two-tailed Mann–Whitney U-test $p$-values: FOXP3 ($p = 1.61\text{E} - 05$), IL7R ($p = 1.53\text{e} - 07$), MS4A1 ($p = 1.00\text{E} - 06$), CD28 ($p = 5.63\text{e} - 06$), IFNG ($p = 4.15\text{e} - 05$). **g** Comparison of baseline RANK metagene and RANKL-treated PyMT tumor acini-derived gene signature between non-responder (NR; $n = 11$) and responder (R; $n = 11$) patients. Significance determined by the two tailed Mann–Whitney U-test. For **a**, **b**: each colored line represents one patient and indicates increase (red), decrease (blue), or no change (black) relative to baseline. Note that all variables were analyzed for all patients, but values for some lines overlap or the indicated population was not detected. Boxplots display median line, IQR boxes, 1.5 × IQR whiskers, and data points. #Responder patients are those with ≥10% increase in TIL infiltration after denosumab treatment. Significance determined by the two-tailed Mann–Whitney U-test.

The strength of our work resides in the fact that two independent studies, a clinical trial and preclinical research on tumor-prone mouse models, equally conclude that the inhibition of RANK signaling increases the anti-tumor immune response and set the basis for additional trials combining denosumab with immunotherapy in presumably immune "cold" luminal BC.

Although the clinical trial primary efficacy endpoint was not met, as tumor cell proliferation was not reduced, a short course of denosumab did induce an increase in the levels of TILs, T and B cells, and CD8$^+$ T-cell infiltration. In contrast with the increased levels of T cells and CD8$^+$ T cells, which were associated with enhanced TIL infiltration, the reduction of Tregs was observed equally in R and NR cases, indicating that it may be driven by additional systemic effects of denosumab, rather than by the loss of RANK signaling in the tumor cells, as suggested by the different results seen in RANK$^{-/-}$ tumors.

Importantly, preclinical genetic mouse approaches evidence that the main immunomodulatory changes induced by denosumab in D-BEYOND—increased in TILs and CD8+ T cells—are replicated when RANK is lost specifically in the tumor compartment. In addition, they add functional relevance to the changes in immune infiltration, as T lymphocytes and CD8+ T cells are responsible for the delayed tumor onset and reduction of tumor-initiating ability observed in RANK-null tumors. In contrast, RANK loss in myeloid cells does not change the tumor immune infiltration. In the PyMT mouse model, the frequency CD8$^+$ T cells also increases after systemic anti-RL treatment and the CD4/CD8 ratio was reduced, but no differences in total leukocyte or lymphocyte infiltration were observed. Differences with the D-BEYOND results might be due to drug-specific aspects, treatment schedule, or tumor divergences.

RANK expression in tumor cells led to a significant increase in the levels of several cytokines and chemokines involved in macrophage and neutrophil recruitment and polarization[28,44,45], in line with the increased infiltration of TAMs and TANs in RANK$^{+/+}$ tumors. Indeed, we found that RANK-expressing human BC cells promote survival of inflammatory neutrophils. Neutrophil depletion significantly delayed tumor appearance in RANK$^{+/+}$, but not in RANK$^{-/-}$ models, supporting a pro-tumorigenic role for neutrophils recruited by RANK$^{+/+}$ tumor cells. Neutrophils have different polarization states and can promote tumorigenesis and metastasis[46]. Our mouse and human data are consistent with the previously reported negative

correlation of TANs and CD8$^+$ T-cell infiltration in NSCLC[47]. Neutrophils have a well-defined role in the suppression of the action of CD8$^+$ T cells[48]. Our results demonstrate that RANK activation in tumor cells increases neutrophil survival and activation inducing an immunosuppressive environment, which could restrict the cytotoxic T-cell response. These findings support the connection between RANK activation in tumor cells, neutrophils, and CD8$^+$ T cells (see Fig. 6).

A critical aspect of current and future clinical trials is the selection of BC patients who may benefit from denosumab treatment, considering the limitations of the RANK IHC. We demonstrate that the RANK metagene we generated, captures RANK activation and predicts the denosumab-driven increase in TILs in BC. Higher RANK metagene, RANK/NF-κB activation in the tumors, and soluble RL at baseline could be better biomarkers than the individual expression levels of RANK or RL for the selection of BC patients who might benefit from denosumab treatment.

The D-BEYOND trial has some limitations, such as the small sample size, the inclusion of only premenopausal patients, and the limited number of triple-negative and HER2$^+$ cases. Whether the immunomodulatory response associated with RL inhibition could also be effective in postmenopausal patients will be addressed in the ongoing trial: D-BIOMARK (NCT03691311). It will be also worth reassessing the clinical outcome of two recent large phase III trials of adjuvant denosumab in early BC, D-CARE, and ABCSG-18, according to the predictive biomarkers we defined as follows: baseline RANK metagene, sRL levels, and the presence of Tregs. The D-CARE study reported no differences in disease-free survival (DFS), whereas the ABCSG-18 trial showed DFS improvement in postmenopausal patients[49–51].

Results in the RANK$^{-/-}$ mouse tumors suggest that upregulation of negative checkpoints and Tregs occurs as a consequence of a proinflammatory, anti-tumor IFNγ-enriched microenvironment[34,52], and may allow RANK$^{-/-}$ tumor cells to evade immune surveillance and grow. The blockade of CTLA4 and PD-1/PD-L1 has revolutionized treatment of highly immunogenic tumors such as melanoma and NSCLC[21,22] but, so far, results in BC have been restricted to basal-like tumors in combination with radiotherapy or chemotherapy[23].

CTLA4 blockade affects mainly the priming phase of the immune response, whereas PD-L1 inhibition works mostly during the effector phase to restore the immune function of previously activated T cells[53]. In both scenarios, we have shown an

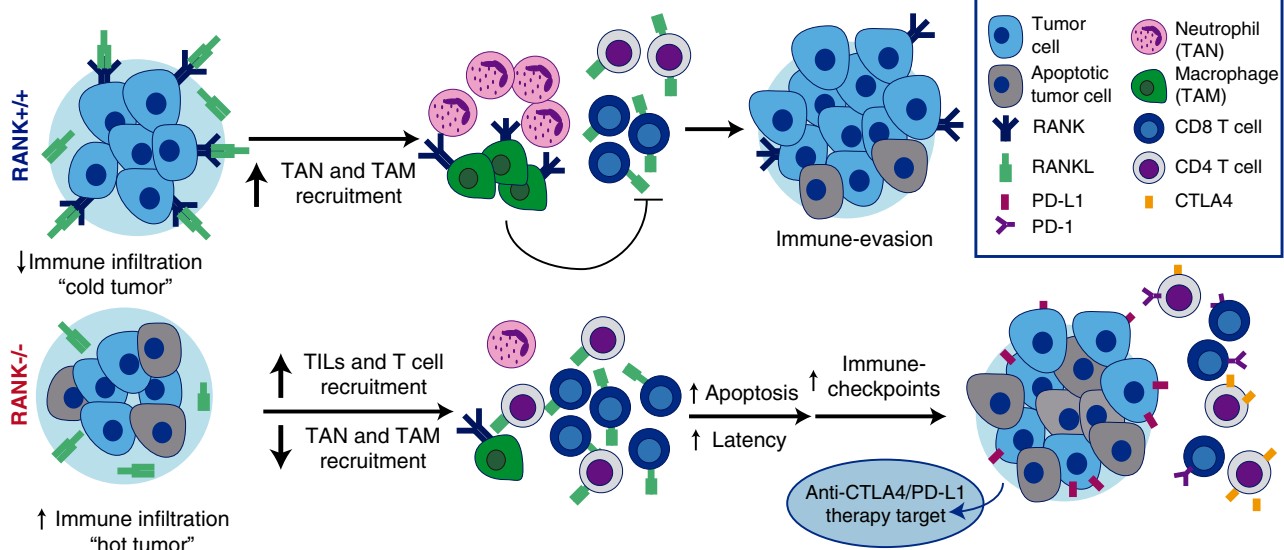

**Fig. 6 The RANK pathway as immune modulator in breast cancer.** RANK expression in luminal breast cancer cells leads to the expression of proinflammatory cytokines/chemokines favoring recruitment of TAMs and TANs, immunosuppressive population that interfere with lymphocyte T-cell recruitment and/or activity. Denosumab (anti-RANKL) or RANK signaling inhibition results in increased TILs, lymphocytes, and CD8+ T-cell infiltration, transforming immune "cold" tumors into "hot" ones and attenuating tumor growth. Eventually, the exacerbated immune response driven by RANK inhibition will induce the expression of immune checkpoints evading immune surveillance and allowing tumor growth. These results support the benefit of combining RANKL and immune-checkpoint inhibitors in luminal breast cancer.

increased benefit after the addition of RL inhibitors to immune checkpoints in RANK$^{+/+}$ tumors, which is highly relevant in poorly immunogenic tumors such as luminal BC. Importantly, the combined treatments show no increased benefit in RANK$^{-/-}$ tumors, indicating that it is driven by the inhibition of RANK signaling in tumor cells. This is a novel mechanism of action, as previous preclinical studies reporting the benefit of the combination were done in melanoma and colon cancer cell lines highly responsive to immunotherapy but lacking RANK expression[54,55]. Although we cannot rule out that denosumab may have additional systemic effects, our findings support that a tumor cell-driven effect contributes to the immunomodulatory effect of denosumab in BC.

The benefit of the combined effect of anti-RL and immune-checkpoint inhibitors will be investigated in the CHARLI trial (NCT03161756), a phase I/II study of the effect of denosumab in combination with nivolumab (an anti-PD-1), with or without ipilimumab (anti-CTLA4), in metastatic melanoma patients, and in the POPCORN trial (ACTRN12618001121257), which will evaluate immune changes in NSCLC patients treated with nivolumab alone or in combination with denosumab. Clinical and preclinical evidence shown in this work encourage the initiation of similar trials in BC.

In summary, compelling clinical and preclinical data reveal an unexpected immunomodulatory role for RANK pathway in luminal early-stage BC and demonstrate denosumab to be a promising agent for enhancing the immune response in luminal BC alone or in combination with immune-checkpoint inhibitors.

## Methods

**Animals and in vivo treatments**. All research involving animals was performed at the IDIBELL animal facility in compliance with protocols approved by the IDI-BELL Committee on Animal Care and following national and European Union regulations. MMTV-PyMT (FVB/N-Tg(MMTV-PyVT)634Mul) were acquired from the Jackson Laboratory[24] and RANK$^{+/-}$ (C57Bl/6) mice from Amgen, Inc.[12]. MMTV-PyMT; RANK$^{-/-}$ mice were obtained by backcrossing the MMTV-PyMT (FvB/N) strain with RANK$^{+/-}$ mice into the C57BL/6 background for at least ten generations. RANKflox/flox (RANK$^{fl/fl}$) were provided by Dr. Joseph Penninger[56] and crossed with either MMTV-PyMT$^{-/+}$ or LysM-cre mice (MGI: 1934631) all in

C57Bl/6 background. The athymic nude *Foxn1$^{nu}$* mice were obtained from Envigo. For RANK depletion in the MMTV-PyMT$^{-/+}$ RANK$^{fl/fl}$ tumors, cells were plated in vitro and infected with lentivirus produced in HEK293T cells. Lentiviral packaging plasmids psPAX2 (Addgene, 12260) and pMD2.G (Addgene, 12259), with either control pLVX-IRES-ZsGreen1 vector (Addgene, 632187), or pLVX-Cre-IRES-ZsGreen1, kindly provided by Dr. Alejandro Vaquero, were used, following Addgene's recommended protocol for lentiviral production. Tumor cells were cultured for 16 h with 1:3 virus-containing medium and, 72 h later, infected cells were FACs-sorted for zsGreen expression before being injected into syngeneic hosts.

RANK-Fc (10 mg/kg, Amgen) was injected subcutaneously three times a week[3,4]. Therapeutic anti-RL (clone IK22/5), anti-CTLA4 (clone 9D9), anti-PD-L1 (clone 10 F.9G2), and isotype control rat IgG2A (clone 2A3) and mouse IgG2b (clone MCP-11) were obtained from BioXCell, and 200 μg were administered intraperitoneally twice per week for treatments starting 72 h after tumor cell injection or three times per week for treatments of established tumors (size > 0.09 cm$^2$). For depletion experiments, anti-CD8 (300 μg, clone 53-5.8), anti-NK1.1 (200 μg, clone PK136), anti-Ly6G (first injection 400 μg, 100 μg thereafter, clone 1A8), and isotype controls mouse IgG2a (clone C1.18.4) and rat IgG1 (clone TNP6A7) were injected intraperitoneally. Treatment was administered on days −1, 0, 3, and 7 after tumor cell injection, and then once per week until experimental endpoint for CD8 and NK depletion. For neutrophil depletion, aLy6G was injected on day −1 and thereafter three times weekly. In all cases, mice were euthanized before tumors exceeded 10 mm in any dimension. Euthanasia was performed by CO$_2$ inhalation. Blood samples were taken flow cytometry analyses to check the depletion 7–10 days and 14–20 days after the first injection. Animals were randomized before beginning the treatment schedule. Mice were kept in individually ventilated and open cages and food and water were provided ad libitum.

**Mouse tumor-cell isolation and tumor-initiation assays**. Draining lymph nodes were removed and fresh tissues were mechanically dissected with a McIlwain tissue chopper and enzymatically digested with appropriate medium (Dulbecco's modified Eagle's medium (DMEM) F-12, 0.3% collagenase A, 2.5 U/mL dispase, 20 mM HEPES, and penicillin–streptomycin 1×) for 40 min at 37 °C. Samples were washed with Leibowitz L15 medium containing 10% fetal bovine serum (FBS) between each step. Erythrocytes were eliminated by treating samples with hypotonic lysis buffer (PAA Laboratories) for 2 min at 37 °C. Single cells were isolated by treating with trypsin (PAA Laboratories) for 2 min at 37 °C. Cell aggregates were removed by filtering the cell suspension with a 70 μm filter and counted. For orthotopic transplants and tumor-limiting dilution assays tumor cells isolated from PyMT;RANK$^{+/+}$ or PyMT; RANK$^{-/-}$ (C57BL/6) mice were mixed 1:1 with Matrigel matrix (Lonza Iberica) and orthotopically implanted in the inguinal mammary gland of 6–10-week-old syngeneic females or *Foxn1$^{nu}$* females. Mammary tumor growth was monitored by palpation and caliper measurements three times per week. Lymph nodes were

treated with hypotonic lysis buffer and then mashed through a 70 μm cell strainer to isolate single cells.

**Flow cytometry**. Single cells from tumors or lymph nodes were resuspended and blocked with phosphate-buffered saline 2% FBS and blocked with FcR blocking reagent (Miltenyi Biotec) for 10 min on ice and incubated for 30 min on ice with the corresponding surface antibodies as follows: CD45-APCCy7 (0.125 μg/mL; 30-F11), CD11b-APC (2.5 μg/mL; M1/70), CD11b-PECy7 (2.5 μg/mL; M1/70), CD8-PE (1 μg/mL; 53-6.7), CD8-FITC (8 μg/mL; 53-6.7), CD4-PE-Cy7 (2 μg/mL; RM4-5), CD25-APC (2 μg/mL; PC61), Ly6C⁻FITC (1.25 μg/mL; HK1.4), Gr1-FITC (2 μg/mL; RB6-8C5), Ly6G-PECy7 (1.25 μg/mL; 1A8), F4/80-PE (1.25 μg/mL; BM8), CD3-PerCPCy5.5 (3.2 μg/mL; 145-2C11), CD3-APC (3.2 μg/mL; 145-2C11), Siglec-F-PerCP-Cy™5.5 (4 μg/mL, E50-2440), CD19-PE (2.5 μg/mL, 6D5), NK1.1-PE (2.5 μg/mL; PK136), PD⁻1-PE (10 μg/mL; 29 F.1A12), PD-L1-PECy7 (1.25 μg/mL; 10 F.9G2), and anti-human CD11b⁻PECy7 (0.8 μg/mL; M1/70) from BioLegend. Apoptosis and necrosis were evaluated using the Annexin V Apoptosis Detection Kit (640930, BioLegend). 7AAD or LIVE/DEAD™ Fixable Green Dead Cell Stain Kit (488 nm) from ThermoFisher was added in the various antibody combinations to remove dead cells. The following antibodies were used for intracellular staining: IFNγ-PE (2 μg/mL; XMG1.2); CTLA4-PerCPCy5.5 (10 μg/mL; UC10-4B9) and CTLA4-PECy7 (5 μg/mL; UC10-4B9) from BioLegend; and FOXP3-FITC (10 μg/mL; FJK-16s) and IL-12-FITC (2 μg/mL; C17.8) from eBioscience. Single-cell suspensions were stimulated in Leibowitz L15 medium containing 10% FBS, 10 ng/mL phorbol 12-myristate 13-acetate (PMA), 1 μg/mL ionomycin, and 5 μg/mL brefeldin A (for IFNγ and CTLA4) or just 5 μg/mL brefeldin A (for IL-12) for 4 h at 37 °C. Surface antibodies were stained first, then fixed with paraformaldehyde (PFA) 4% (in the case of cytokines) or Fixation Reagent of the Foxp3/Transcription Factor Staining Buffer Set from eBioscience (in the case of FOXP3), and permeabilized using Permeabilization Buffer of the Foxp3/ Transcription Factor Staining Buffer Set from eBioscience. The intracellular proteins were then stained. FACS analysis was performed using FACS Canto and Diva software. Cells were sorted using MoFlo (Beckman Coulter) at 25 psi with a 100 μm tip.

Blood samples were collected in tubes containing heparin and stained with CD45-APC-Cy7 (0.125 μg/mL; 30-F11), CD11b-PECy7 (2.5 μg/mL; M1/70), CD3-PerCPCy5.5 (3.2 μg/mL; 145-2C11), CD8-PE (1 μg/mL; 53-6.7), NK1.1-PE (2.5 μg/mL; PK136), Ly6G-PECy7 (1.25 μg/mL; 1A8), and Gr1-FITC (2 μg/mL; RB6-8C5) for 30 min at room temperature (RT) in the dark. Versalyse (Beckman Coulter) containing 0.1% PFA was added to the samples and incubated for 10 min at RT in the dark before passing them through the cytometer.

**IHC in mouse tumor tissues**. Mouse tissue samples were fixed in formalin and embedded in paraffin. Three-micrometer sections were cut for histological analysis and stained with hematoxylin and eosin (HE). Three-micrometer tissue sections were used for immunostaining. Primary antibody was incubated overnight at 4 °C, detected with biotinylated secondary antibodies and streptavidin horseradish peroxidase (Vector), and revealed with DAB substrate (DAKO). CD3 and CD8 immunostaining was performed in the Histopathology Core Unit of the Spanish National Cancer Research Centre (CNIO, Madrid, Spain), using antibodies CD3 (clone M20 from Santa Cruz Biotechnology) and CD8 (clone 94 A from the Monoclonal Antibodies Core Unit of the CNIO). For RANK IHC, antigen retrieval was performed with Protease XXIV at 5 U/ml for 5 min (P8038, Sigma) and the anti-RANK (R&D AF692, 1:200).

**Real-time PCR**. Total RNA was extracted with Tripure Isolation Reagent (Roche) or Maxwell RSC Simply RNA Tissue kit (AS1340, Promega). Frozen tumor tissues were fractionated using glass beads (Sigma-Aldrich) and the PrecCellys® 24 tissue homogenizer (Berting Technologies), and Polytron PT 1200e (Kinematica). cDNA was produced by reverse transcription using 1 μg of RNA in a 35 μL reaction with random hexamers following the kit instructions (Applied Biosystems). In the case of sorted cells, RNA was retrotranscribed with Superscript II Reverse Transcriptase in a 20 μL reaction carried out according to the manufacturer's instructions (ThermoFisher). cDNA (20 ng/well) for whole tumors were analyzed by SYBR green real-time PCR with 10 μM primers using a LightCycler® 480 thermocycler (Roche). Analyses were performed in triplicate. *Hprt1* was used as the reference gene. The following primer pairs were used for each gene: *Hprt1*, 5′-TCAGT-CAACGGGGGACATAAA-3′, 5′- GGGGCTGTACTGCTTAACCAG-3′; *Prf1*, 5′-CTGGATGTGAACCCTAGGCC-3′, 5′-GCGAAAACTGTACATGCGAC-3′; *Ifnγ*, 5′-CACGGCACAGTCATTGAAAG-3′, 5′-CCATCCTTTTGCCAGTTCCTC-3′; *Il-1β*, 5′-CAACCAACAAGTGATATTCTCCATG-3′, 5′-GATCCA-CACTCTCCAGCTGCA-3′; *Casp4*, 5′-AATTGCCACTGTCCAGGTCT-3′, 5′-CTCTGCACAACTGGGGTTTT-3′; *S100a9*, 5′-TCAGACAAATGGTGGAAGCA-3′, 5′-GTCCTGGTTTGTGTCCAGGT-3′.

For human cell line samples, the following primer sequences were used: *PPIA*, 5′-GGGCCTGGATACCAAGAAGT-3′, 5′-TCTGCTGTCTTTGGGACCTT-3′; *BIRC3*, 5′-GGTAACAGTGATGATGTCAAATG-3′, 5′-TAACTGGCTTGAACTTGACG-3′; *ICAM1*, 5′-AACTGACACCTTTGTTAGCCACCTC-3′, 5′-CCCAGTGAAATGCAAACAGGAC-3′; *NFkB2*, 5′-

GGCGGGCGTCTAAAATTCTG-3′, 5′-CCAGACCTGGGTTGTAGCA-3′; *RELB*, 5′-TGTGGTGAGGATCTGCTTCCAG-3′, 5′-TCGGCAAATCCGCAGCTCTGAT-3′.

**Mouse RNA labeling and hybridization to Agilent microarrays**. Hybridization to the SurePrint G3 Mouse Gene Expression Microarray (ID G4852A, Agilent Technologies) was conducted following the manufacturer's two-color protocol (Two-Color Microarray-Based Gene Expression Analysis v. 6.5, Agilent Technologies). Dye swaps (Cy3 and Cy5) were performed on RNA amplified from each sample. Microarray chips were then washed and immediately scanned using a DNA Microarray Scanner (Model G2505C, Agilent Technologies).

**Tumor acinar cultures and cytokine array**. Isolated tumor cells coming from RANK$^{+/+}$ or RANK$^{-/-}$ transplants were seeded on top of growth factor-reduced matrigel (one million cells/well in six-well plates) in growth medium (DMEM-F-12, 5% FBS, 10 ng/mL of epidermal growth factor (EGF), 100 ng/mL cholerin toxin, 5 μg/mL insulin and 1x penicillin/streptomycin).

For cytokine arrays, tumor supernatants were collected 72 h after plating. A pool of three supernatants derived from three independent tumor transplants and primary tumors was used for the analyses. Multiplex quantification of cytokines and chemokines of supernatants collected from 3D acinar cultures was performed using the Mouse Cytokine Array C1000 (RayBiotech) following the manufacturer's instruction and using the recommended ImageJ plug-in. To detect genes affected by RANK activation, 1 μg/mL RL was added 24 h after tumor plating. RNA was extracted 24 h after RL stimulation for hybridization to a gene expression microarray, as previously described.

**Cell line culture and lentiviral transduction**. The human BC cell lines MCF7 and HCC1954 were purchased from the American Type Culture Collection (ATCC). ATCC provides molecular authentication in support of their collection through their genomics, immunology, and proteomic cores, as described, by using DNA barcoding and species identification, quantitative gene expression, and transcriptomic analyses (ATCC Bulletin, 2010). Cells were grown in DMEM and RPMI 1640 medium, respectively, supplemented with 10% FBS and 1% penicillin–streptomycin solution (all from Gibco). The cells were grown at 37 °C in the presence of 5% $CO_2$ in humidified incubators and were tested for the absence of mycoplasma.

To ectopically express green fluorescent protein (control) or RANK (*TNFRSF11A*), the corresponding genes were cloned in the lentiviral vector pSD-69 (PGK promoter, generously donated by S Duss and M Bentires-Alj) following Gateway cloning protocols. To knock down the expression of endogenous RANK, we used the lentiviral vector pGIPZ clones V3LHS_307325 and V3LHS_400741 with RANK-specific short hairpin RNA expression (Dharmacon). As a control (ctrl), we used a verified non-targeting clone (Dharmacon). Lentiviruses were prepared in HEK293T cells with packaging and envelope plasmids psPAX2 and pMD2.G (AddGene). Transduced cells were selected with 1.5 μg/ml puromycin, starting 3 days after infection.

**Human neutrophil and T-cell isolation and culture**. Peripheral blood was provided by the "Banc de Sang I Teixits" (Hospital Universitari de Bellvitge). Mononuclear cells were isolated from buffy coats using Ficoll-plus gradient (GE Healthcare Bio-Sciences). Neutrophils were isolated from the red fraction, then purified by dextran sedimentation. Purified cells were resuspended at $5 \times 10^6$ cells/ mL in RPMI supplemented with 10% of FBS and 50 U/mL streptomycin and penicillin. FACS analysis was performed to detect CD66b (G10F5, BD Bioscience) to confirm purity (98% average).

Neutrophil apoptosis and activation were analyzed culturing $10^4$ neutrophils per well in 96-well plates over 24 h in the indicated medium or CM. Apoptosis was measured using the Annexin AV Apoptosis Detection Kit (640930, BioLegend) and activation was detected by staining for CD11b following the previously described flow cytometry staining protocol.

**Clinical trial design and patient characteristics**. Twenty-seven patients were enrolled in the D-BEYOND trial: the first patient enrolled on 2 October 2013 and the last patient enrolled on 9 June 2016. D-BEYOND was a prospective, single-arm, multi-center, open label, pre-operative "window-of-opportunity" phase IIa trial (NCT01864798). All patients received two injections of denosumab 120 mg subcutaneously, administered 7–12 days apart, prior to surgical intervention. Surgery was performed 10–21 days after the first dose of denosumab (median, 13 days). Post-study treatment was at the discretion of the investigator. Snap-frozen and formalin-fixed, paraffin-embedded (FFPE) tumor and normal tissues were collected at baseline (pretreatment) and at surgery (posttreatment). Normal tissues (snap-frozen and FFPE) were defined as being at least 1 cm away from tumor, another quadrant, or contralateral breast biopsies. All samples (including normal) were reviewed by a pathologist to assess epithelial content. Eligible patients were premenopausal women with histologically confirmed newly diagnosed operable primary invasive carcinoma of the breast, who had not undergone previous treatment for invasive BC. Other key eligibility criteria included a tumor size > 1.5 cm, any nodal status, and known ER, progesterone receptor (PR), and human epidermal

growth factor receptor 2 (HER2) status. Key exclusion criteria included bilateral invasive tumors, current or previous osteonecrosis, or osteomyelitis of the jaw, and known hypersensitivity to denosumab. Evaluation of conventional BC markers including ER, PR, HER2, and Ki-67 were centrally performed at the Institut Jules Bordet (IJB). ER and PR status were defined according to the American Society of Clinical Oncology and the College of American Pathologists (ASCO-CAP) guidelines. BC subtypes were defined according to the St Gallen 2015 Consensus Meetings[57] using immunohistochemical surrogates as follows: Luminal A: ER and/or PR(+), HER2(−), Ki-67 < 20%; Luminal B: ER and/or PR(+), HER2(−), Ki-67 ≥ 20; Basal: ER(−), PR(−), and HER2(−), irrespective of Ki-67 score; and HER2: HER2(+), irrespective of ER, PR, or Ki-67. All 4 HER2+ patients included in the study were ER+ PR+. The full study protocol is available as Supplementary Note 1 in the Supplementary Information file.

Serious and non-serious AEs were collected from the day of signed informed consent until one month after the final administration of the study drug, except for the project-specific AEs, for which the reporting was extended to 3 months after the final dose of denosumab. Safety data were evaluated using the National Cancer Institute Common Terminology Criteria for Adverse Events (NCI-CTCAE v 4.0). AEs were coded according to the Medical Dictionary for Regulatory Activities (version 20.1). All non-serious AEs are summarized in Supplementary Data 6, the most frequent one being arthralgia (4/27, 14.8%). This study was approved by the Ethics Committee of the trial sponsor; the Medical Ethics Committee of the Institute Jules Bordet (IJB No.: 2064) and and the Melbourne Health Human Research Ethics Committee. All patients provided written informed consent prior to study entry.

One patient was excluded because she had a ductal in situ carcinoma and two patients were excluded because of lack of available tumor tissue. Another patient was excluded from TIL evaluation due to tissue exhaustion. The primary study endpoint was a GM decrease in the percentage of Ki-67-positive cells assessed by IHC. Key secondary endpoints included absolute Ki-67 responders (defined as <2.7% Ki-67 IHC staining in the posttreatment tumor tissue), decrease in serum C-terminal telopeptide (CTX) levels measured by enzyme-linked immunosorbent assay (ELISA), increase in apoptosis as detected by cleaved caspase-3 or terminal deoxynucleotidyl transferase dUTP nick end labeling assays, evaluate the tolerability of a short course of denosumab, and observe changes in TIL percentage in tumor tissue evaluated on HE slides. Changes in the infiltration of immune populations as measured by IHC were also performed. Paired samples of breast tumor and normal tissue at baseline and at surgery were required. The limited epithelial content precluded analyses of changes in the paired normal tissues. Gene expression analyses in paired tumor and normal tissue at baseline and at surgery was performed for patients with enough epithelial content. Additional secondary endpoints include: change in RANK/RL gene expression and signaling, change in tumor proliferation rates using gene expression, change in expression levels from genes corresponding to mammary progenitor populations, estrogen pathways, immune pathways, and gene expression changes in the paired samples of surrounding normal tissue when available. All primary, secondary, and exploratory endpoints performed are summarized in Supplementary Data 17.

**Enzyme-linked immunosorbent assay.** Serum concentrations of human sRL were centrally assessed at IJB in triplicate, using an ELISA according to the manufacturer's instructions (Biomedica, Austria). sRL bound to denosumab is not be detected by this assay. Serum CTX levels were routinely evaluated in each center by ELISA.

**Pathological assessment and immunohistochemical staining of human tumor samples.** Tumor cellularity was centrally assessed on HE-stained tissue sections from FFPE and frozen human tumor samples. For patients with multiple samples, the sample with the highest tumor content was chosen for further analyses. The percentage of intratumoral and sTILs was independently evaluated by two trained pathologists (R.S. and G.V.D.E.) who were blinded to the clinical and experimental data on the HE slides, following the International TIL Working Group 2014 methodology, as described elsewhere[58]. Median tumor cellularity ranged between 35% and 90%. TIL proliferation was assessed as the percentage of Ki-67+ TILs among all TILs.

Tissue sections (4 μm) from FFPE tissues of human primary breast tissue were used to assess RANK and RL. For each patient, representative unstained slides of the primary tumor were shipped to NeoGenomics Laboratories (California, USA) for immunohistochemical staining of RANK (N1H8, Amgen), RL (M366, Amgen), blinded to clinical information. The percentage of stained cells and their intensity (0, negative; 1+, weak; 2+, moderate; and 3+, strong) were recorded as described previously[23].

An H-score was calculated using the following formula: H = (% of cells of weak intensity × 1) + (% of cells with moderate staining × 2) + (% of cells of strong staining × 3). The maximum possible H-score is 300, corresponding to 100% of cells with strong intensity.

Serial FFPE tissue sections (4 μm) were immunohistochemically stained for CD3/CD20, CD4/CD8, and FOXP3/CD4 dual staining, as well as single Ki-67 and cleaved caspase-3 staining on a Ventana Benchmark XT automated staining instrument (Ventana Medical Systems)[59]. The antibodies used for dual IHC are as follows: CD3 (IR503, polyclonal), CD8 (C8/144B, IR623), and CD20 (L26, IR604)

from Dako; CD4 (RBT-CD4, BSB5150) from BioSB; FOXP3 (236 A/E7, 14-4777-82) from E-Bioscience; Ki-67 (Clone MIB-1) from Dako; and cleaved caspase-3 (ab2302) from Abcam. T cells were quantified by CD3 protein expression, B cells by CD20 protein expression, cytotoxic T cells by CD4-negative and CD8 -positive expression, and Treg cells by simultaneous CD4 and FOXP3 expression. Scoring was defined as the percentage of immune-positive cells among stromal and tumoral area.

For multiplex IHC, FFPE tissue sections (4 μm) were processed manually. Briefly, slides were heated at 37 °C overnight, deparaffinized, and then fixed in neutral-buffered 10% formalin. The presence of helper T cells (CD4), cytotoxic T cells (CD8), B cells (CD20), Tregs (FOXP3), macrophages (CD68), cancer cells (pan-cytokeratin), and cell nuclei (4′,6-diamidino-2-phenylindole) was assessed using a serial same-species fluorescence-labeling approach that employs tyramide signal amplification and microwave-based antigen retrieval and antibody stripping in accordance with the manufacturer's instructions (Opal Multiplex IHC, Perkin Elmer). Staining was visualized on a Zeiss LSM 710 confocal microscope equipped with PMT spectral 34-Channel QUASAR (Carl Zeiss). All IHC slides were centrally reviewed by a breast pathologist (R.S.).

**RNA extraction from human samples and RNA-seq.** RNA was extracted from frozen tumor and normal tissue using the AllPrep DNA/RNA Mini kit (Qiagen, Germany) according to the manufacturer's instructions. RNA quality was assessed using a Bioanalyzer 2100 (Agilent Technologies). A total of 22 patients had sufficient tumor RNA quantity from both pre- and posttreatment timepoints. A total of 11 patients had sufficient RNA quantity in normal tissue samples from both pre- and posttreatment timepoints. Among the patients without enough RNA quantity in normal tissue, six had biopsies containing mainly fatty tissue without any epithelial cell. Indexed cDNA libraries were obtained using the TruSeq Stranded Total RNA Kit (Illumina) following the manufacturer's recommendations. The multiplexed libraries were loaded onto a NovaSeq 6000 apparatus (Illumina) using a S2 flow cell and sequences were produced using a 200 Cycle Kit (Illumina).

**Bioinformatic analyses.** RNA-seq read pairs from the D-BEYOND samples were trimmed using Trimmomatic[60]. Alignment was performed using STAR[10]. The number of reads mapping to each gene was assessed with the Rsamtools package in the R environment. As gene expression profiles of tissues taken at biopsy and surgery are known to be sensitive to differences in tissue-handling procedures[61], we used a publicly available dataset from the no-treatment arm of POETIC study to filter-out differentially expressed genes. This study included 57 pairs of samples from untreated patients taken at diagnosis (baseline) and surgery (GEO ID: GSE73235[61]). We filtered out 3270/21.931 (14.9%) genes that were differentially expressed between diagnosis and surgery by using a strict cutoff of a raw value of $P < 0.05$ from a non-parametric Mann–Whitney $U$-test. Differential expression was analyzed with DESeq2 v.1.14.1R/Bioconductor package[62] using raw count data. Significantly differentially expressed genes were selected if they had a qval of <0.05 and an absolute log2-fold change of >0.5. We used the GAGE v.2.24.0 R/Bioconductor package[63] to identify significantly enriched biological processes from the Biological Process from GO database. CIBERSORT software was used[40] to refine the subsets of immune cells present in each sample. Reads per kilobase of transcript, per million mapped reads expression data were uploaded to www.cibersort.standford.edu and CIBERSORT was run using LM22 as a reference matrix and, as recommended for RNA-seq data, quantile normalization was disabled.

All other parameters were set to default values. Output files were downloaded as tab-delimited text files and immune cell subsets that were present in fewer than ten samples were discarded.

We reported the ten aggregates as described before [PMID: 29628290]:

T.cells.CD8 = T.cells.CD8,

T.cells.CD4 = T..CD4.naive + T..CD4.memory.resting + T..CD4.memory.activated,

T.reg = T.cells.regulatory..Tregs.

B.cells = B.cells.naive + B.cells.memory,

NK.cells = NK.cells.resting + NK.cells.activated,

Macrophage = Macrophages.M0 + Macrophages.M1 + Macrophages.M2,

Dendritic.cells = Dendritic.cells.resting + Dendritic.cells.activated,

Mast.cells = Mast.cells.restin + Mast.cells.activated,

Neutrophils = Neutrophils,

Eosinophils = Eosinophils

RNA-seq data have been deposited under EGA accession number EGAS00001003252 as a fatsq file (available on request from the IJB Data Access Committee).

The prototype-based co-expression module score for *TNFRSF11A* (RANK metagene) and *TNFSF11* (RL metagene) was computed for each sample as Modulescore = $\sum_{i=1}^{100} w_i x_i$. Where $x_i$ is the expression of the top 100 genes positively correlated with *TNFRSF11A* or *TNFSF11* at baseline (before treatment) and $w_i$ is the Pearson's correlation coefficient between $x_i$ and *TNFRSF11A* or *TNFSF11*.

The public signatures of RANK/NFκB were retrieved from MSigDB[64] (Cell Systems, PMID:26771021) and computed using the GM and then scaling. RL-induced genes in mouse MECs (WT and *Rank* overexpression) were retrieved from publically available GEO dataset: GSE66174.

Mouse microarray data were feature-extracted using Agilent's Feature Extraction Software (v. 10.7), using the default variable values.

Outlier features in the arrays were flagged by the same software package. Data were analyzed using the *Bioconductor* package in the R environment. Data preprocessing and differential expression analysis were performed using the *limma* and *RankProd* packages, and the most recently available gene annotations were used. Raw feature intensities were background-corrected using the *normexp* background-correction algorithm. Within-array normalization was done using spatial and intensity-dependent *loess*. *Aquantile* normalization was used to normalize between arrays. The expression of each gene was reported as the base 2 logarithm of the ratio of the value obtained for each condition relative to the control condition. A gene was considered differentially expressed if it displayed a *pfp* (proportion of false positives) < 0.05, as determined by a non-parametric test.

**Statistical analyses**. All statistical tests comparing pre- and posttreatment paired values were done using the sign test or Student's paired samples *t*-test. All IHC values were log-transformed to give values of $\log_{10}(x + 1)$, thereby overcoming the problem of some raw variable values being zero. To compare NRs and responders, the Mann–Whitney *U* and Fisher's exact tests were used for continuous and categorical variables, respectively. All correlations were measured using the Spearman's non-parametric *rho* coefficient. All reported *P*-values were two-tailed. All analyses were performed using R version 3.3.3 (available at www.r-project.org) and Bioconductor version 3.6. No correction was made for multiple testing for exploratory analyses, except for the gene expression analysis, for which the false discovery rate was used.

Mouse experimental data were analyzed using GraphPad Prism version 5. Differences were analyzed with a two-tailed Student's *t*-test, an F-test, or an unpaired-samples *t*-test against a reference value of 1. Tumor growth curves were compared using two-way analysis of variance. Frequency of tumor initiation was estimated using the extreme limiting dilution assay (http://bioinf.wehi.edu.au/software/elda/). Regression analysis of the growth curves' mean for the anti-CTLA4, anti-RL, and anti-PD-L1 treatments was performed, and $2 \times 2$ $\chi^2$-contingency tables (two-tailed probabilities) were used to evaluate responses. The statistical significance of group differences is expressed by asterisks: *$p < 0.05$; **$p < 0.01$; ***$p < 0.0001$; ****$p < 0.0001$.

## Data availabiliy

Raw microarray data from preclinical samples have been deposited in GEO, access number GSE119464, and are publicly available. Patients' RNA-seq data have been deposited under EGA accession number EGAS00001003252. Access can be obtained by contacting the Institute Jules Bordet Data Access committee or Christos Sotiriou (christos.sotiriou@bordet.be). Raw clinical data are available as Supplementary Data 18. The full study protocol is available in the Supplementary Information file. The POETIC clinical trial gene expression data used in this study are available in the GEO database under accession code GSE73235. MECsWT and Rank overexpression microarray data used in this study are available in the GEO database under accession code GSE66174. The remaining data are available within the Article, Supplementary Information, or available from the authors upon request.

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

## Acknowledgements

We thank the patients who contributed to this study and acknowledge the clinical staff for their dedication. The D-BEYOND clinical trial was sponsored by Jules Bordet Institute, which was responsible for the management of the study. This work was supported by grants to E. González-Suárez by MICINN (SAF2014-55997-R, SAF2017-86117-R) co-funded by FEDER funds/European Regional Development Fund (ERDF)—a way to build Europe), the European Research Council (ERC) under the European Union's Horizon 2020 research and innovation program (grant agreement No 682935), and Fundació La Marató de TV3. We thank CERCA Programme/Generalitat de Catalunya for institutional support. P.P. and S.B. were and A.B. is recipient of a predoctoral grant from the MICINN. We are grateful to William C. Dougall and AMGEN, Inc. for supporting the design of the D-BEYOND trial and providing RANKL, RANK-Fc reagents, and RANK⁻/⁻ mice. We thank the IDIBELL Animal Facility for their assistance with mouse colonies, Esther Castaño and the scientific services of the University of Barcelona for their assistance with FACS analyses, and P. Gonzalez-Santamaria, G Perez-Chacon, and M Jimenez for critical reading of the manuscript. C.S. and B.N. are supported by the National Fund for Research (FNRS) and Televie. R.S. is supported by a grant from the Breast Cancer Research Foundation (BCRF), grant number 17-194. This work was also supported in part by the Cancer Center Support Grant of the National Institutes of Health (Grant Number P30CA008748). We thank Samira Majjaj and Delphine Vincent for technical assistance. We extend gratitude to the patients who participated in the D-BEYOND study. This clinical study has been supported by research funding from Amgen.

## Author contributions

G.Y. and C.G.A.: Collection and/or assembly of data, data analysis and interpretation, and manuscript writing. M.Z., P.P., E.M.T., S.B., M.C., A.B., and E.H.: Collection and/or assembly of data, and data analysis and interpretation. T.W. and L.P.: Conception and design, and data analysis and interpretation. P.M.: data analysis and interpretation. E.G.-S.: Conception and design of the preclinical study, financial support, collection and/or assembly of data, data analysis and interpretation, and manuscript writing. M.P., C.S., S.L., and H.A.A.: Conception and design of the D-BEYOND clinical trial. B.N.: Collection and/or assembly of clinical trial and biological samples data, data analysis, and manuscript writing. F.R.: Experiments on patients' samples supervision, data analysis and interpretation, and final approval of manuscript. S.M. and D.V.: Statistical design and clinical trial data analysis. M.M.: collection and/or assembly of the clinical trial data and biological samples. R.S., D.L., and G.V.E.: Pathology assessment of the biological samples. P.V., L.P., H.W., P.S., G.L., and J.P.: Patient consent and recruitment. S.G. and K.W.-G.: Immunological assessment of the biological samples. C.V.: Clinical trial project management. All: interpretation of the data analysis and final approval of the manuscript.

## Competing interests

R.S. reports non-financial support from Merck and Bristol Myers Squibb; research support from Merck, Puma Biotechnology, and Roche; and personal fees from BMS for an advisory board meeting and from Roche for an advisory board related to a trial-research project. H.A.A. is advisory board at Roche and current employee of Innate Pharma. E.G.S and G.J.L. have served on advisory boards for Amgen and has received honoraria and research funding from Amgen. S.L. receives research funding from Novartis, Merck, BMS, Roche-Genentech, Puma Biotechnology, Pfizer, and uncompensated advisory board of Novartis, Merck, BMS, Roche-Genentech, Puma Biotechnology, Pfizer, and Seattle Genetics. The remaining authors declare no competing interests.

## Additional information

Clara Gómez-Aleza [1,20], Bastien Nguyen [2,3,20], Guillermo Yoldi[1,20], Marina Ciscar[1,4], Alexandra Barranco[1,4], Enrique Hernández-Jiménez [1], Marion Maetens[2], Roberto Salgado[2,5], Maria Zafeiroglou[1], Pasquale Pellegrini[1], David Venet[2], Soizic Garaud[6], Eva M. Trinidad [1], Sandra Benítez [1], Peter Vuylsteke[7], Laura Polastro[8], Hans Wildiers [9], Philippe Simon[10], Geoffrey Lindeman [11], Denis Larsimont[12], Gert Van den Eynden[6], Chloé Velghe[13], Françoise Rothé[2], Karen Willard-Gallo [6], Stefan Michiels [14], Purificación Muñoz[1], Thierry Walzer[15], Lourdes Planelles[16], Josef Penninger[17,18], Hatem A. Azim Jr[19], Sherene Loi [11], Martine Piccart[8], Christos Sotiriou [2,8,21✉] & Eva González-Suárez [1,4,21✉]

[1]Oncobell, Bellvitge Biomedical Research Institute, IDIBELL, Barcelona, Spain. [2]Breast Cancer Translational Research Laboratory J.-C. Heuson, Institut Jules Bordet, Université Libre de Bruxelles, Brussels, Belgium. [3]Marie-Josée and Henry R. Kravis Center for Molecular Oncology, Memorial Sloan Kettering Cancer Center, New York, NY 10065, USA. [4]Molecular Oncology, Spanish National Cancer Research Centre (CNIO), Madrid, Spain. [5]Department of Pathology, GZA-ZNA Ziekenhuizen, Antwerp, Belgium. [6]Molecular Immunology Unit, Institut Jules Bordet, Université Libre de Bruxelles, Brussels, Belgium. [7]Department of Medical Oncology, Université Catholique de Louvain, CHU UCL, Namur, site Sainte-Elisabeth, Namur, Belgium. [8]Department of Medical Oncology, Institut Jules Bordet, Université Libre de Bruxelles, Brussels, Belgium. [9]Department of Oncology, KU Leuven-University of Leuven, Leuven, Belgium. [10]Department of Obstetrics and Gynaecology, Erasme, Université Libre de Bruxelles, Brussels, Belgium. [11]Peter MacCallum Cancer Centre, The Walter and Eliza Hall Institute of Medical Research and The Royal Melbourne Hospital, Melbourne, VIC, Australia. [12]Department of Pathology, Institut Jules Bordet, Université Libre de Bruxelles, Brussels, Belgium. [13]Clinical Trial Supporting Unit, Institut Jules Bordet, Université Libre de Bruxelles, Brussels, Belgium. [14]Service de Biostatistique et D'Epidémiologie, Gustave Roussy, CESP, U1018, Université Paris-Sud, Faculté de Médcine, Université Paris-Saclay, Villejuif, France. [15]Centre International de Recherche en Infectiologie, CIRI, Inserm U1111, CNRS, Université Claude Bernard, Lyon, France. [16]BiOncotech Therapeutics, Parc Cientific Universitat, Valencia, Spain. [17]Life Sciences Institute, University of British Columbia, Vancouver, BC, Canada. [18]IMBA, Institute of Molecular Biotechnology of the Austrian Academy of Sciences, Vienna, Austria. [19]Division of Hematology/Oncology, Department of Medicine, American University of Beirut, Beirut, Lebanon. [20]These authors contributed equally: Clara Gómez Aleza, Bastien Nguyen, Guillermo Yoldi. [21]These authors jointly supervised this work: Christos Sotiriou, Eva González-Suárez. ✉email: christos.sotiriou@bordet.be; egonzalez@cnio.es

