## [Peer Review File · Nature Communications]

REVIEWER COMMENTS

Reviewer #1 (Remarks to the Author):

The authors have, in general, satisfactorily addressed the concerns. Key new data strengthening the paper include additional controls (analyses in RANK^{-/-} tumor cells), additional supportive data (RANK metagene analysis), and additional mechanistic data (mouse depletion experiments, human peripheral blood neutrophil experiments) consistent with a role for neutrophils in the suppressive microenvironment mediated by RANK signaling.

Reviewer #2 (Remarks to the Author):

Manuscript NCOMMS-20-23201A

Title: "Inhibition of RANK signaling in breast cancer induces an anti-tumor immune response orchestrated by CD8⁺ T cells"

Authors: Yoldi et al

Summary

In this revised manuscript, Yoldi et al investigate the role of RANK signaling in breast cancer cells on the immune cell composition and immunotherapy response of breast tumors. The revised manuscript addresses many of the suggestions and concerns raised by the Referees, and the message of the manuscript is clearly strengthened by the newly added experiments. In particular, the authors have added new experiments to confirm in a more controlled setting that RANK expression in cancer cells influences the immune cell composition of tumors, and they now show more clearly that the synergy between RANK inhibition and immune checkpoint blockade depends on RANK expression in tumors. Newly added in vitro studies reveal that RANK pathway activation in human breast cancer cells enhances the survival and activation of cultured neutrophils. In addition, the authors have performed additional analyses that reinforce the link between the mouse and human studies.

The main conclusion of the authors is that RANK expression in cancer cells promotes an immunosuppressive microenvironment and that targeting RANK pathway may represent a new strategy to enhance an anti-tumor immune response in 'cold' breast cancers.

General comments:

The strength and novelty of this manuscript has been improved by the newly added experiments. The authors have addressed most of my concerns. One of our concerns is not fully addressed, namely the (molecular) mechanisms by which RANK signaling in cancer cells increases TANs and TAMs which restrict T cell immunity. But we appreciate that the authors have strengthened the functional crosstalk between RANK expressing tumor cells, neutrophils and CD8⁺ T cells. We understand that a full dissection of the underlying (soluble) mediators involved in this crosstalk is complex, and not feasible at this stage.

There are a few remaining minor points that need (textual) clarification:

Page 27, lines 606-614: the serial transplantation experiment needs more explanation. For the reader, it is not clear what the exact purpose is of the serial transplants. In the rebuttal, the authors refer to their previous publication, however, this should also be clarified in the current manuscript. It is not clear why the authors consider the second transplant as a 'adjuvant' therapy setting, as normally adjuvant therapy implies a treatment after surgical removal of a tumor. But in the experimental set-up, the tumor is implanted in a new mouse, which does not truly mimic a

surgical removal setting. The added value of this experiment should be better explained.

Page 28, lines 644-645: the authors state that s100a9 mRNA expression is higher in RANK+/- tumors, but the statistical analysis does not support this conclusion. The authors should rephrase this.

Sup Figure S1a: wrong legends in x- or y- axis of third facs plot (SSC-A is indicated in both axes).

Sup Figure S1c: in the legends the authors wrote that each dot represents one tumor transplants derived from 12-13 primary tumors, but in the figure, there are more than 12-13 dots. The authors should correct the sentence or explain the discrepancy.

Reviewer #3 (Remarks to the Author):

The authors have made suitable changes that address all of my previous concerns. I am also impressed with their extensive and assiduous efforts to address all of the concerns raised by the other 2 reviewers with the inclusion of new data. The point that mouse GEMMs are important models for the purposes used here to overcome many issues of patient tumour heterogeneity is also a key and valid one.

Reviewer #4 (Remarks to the Author):

The paper covers comprehensive analysis from mice and cell line experiments to human clinical trial. The revised version included additional analyses to improve the paper. Specifically, the control arm in Poetic trial was used to filter out differentially expressed genes due to random sampling differences. Such strategy was justified because both trials have comparable study cohort with majority breast cancer patients in Lum A and B. Bioinformatics analysis was also used to develop RANK metagene and RANKL metagene to compensate IHC limitations of RANK protein detection and to improve selection of tumor-specific pharmacodynamic biomarkers. Analysis results were enhanced by showing high association of RANK metagene with multiple RANK/NFKB pathways and with immuno-response.

A few minor comments:

1. The authors plan to use the ongoing trial, D-BIOMARK (NCT03691311), to validate RANKL inhibition associated immunomodulatory response. However, the clinical trial Gov. website (<https://clinicaltrials.gov/ct2/show/NCT03691311>) indicates change of Ki67 and Cleaved Caspase-3 as the primary endpoints and ratio of RANK/RANKL as secondary endpoint. It is unclear how the D-BIOMARK is capable for validation purpose with sufficient statistical power.
2. While plot for comparing RankL metagene between R vs NR was reported in Figure S6, the plot for Rank metagene was missing. Rank metagene was shown to have good correlation with multiple RANK/NFKb pathways. It would be informative to know how many genes in the Rank metagene were overlapped with the correlated pathways. Another question is: Figure S6a had RankL score higher in NR, but Figure 5e had sRankL higher in R group. Any explanation.
3. It would be more informative to report immune-response in the Table 1 of clinicopathological features of the 24 evaluable patients.
4. The study showed rank metagene as a potential biological endpoint. It is unclear if it is associated with clinical endpoints.

Point-by-point response

Reviewer #1 (Remarks to the Author):

The authors have, in general, satisfactorily addressed the concerns. Key new data strengthening the paper include additional controls (analyses in RANK^{-/-} tumor cells), additional supportive data (RANK metagene analysis), and additional mechanistic data (mouse depletion experiments, human peripheral blood neutrophil experiments) consistent with a role for neutrophils in the suppressive microenvironment mediated by RANK signaling.

We thank Reviewer #1 for the feedback to improve the manuscript and her/his positive comments.

Reviewer #2 (Remarks to the Author):

Manuscript NCOMMS-20-23201A

Title: "Inhibition of RANK signaling in breast cancer induces an anti-tumor immune response orchestrated by CD8⁺ T cells"

Authors: Yoldi et al

Summary

In this revised manuscript, Yoldi et al investigate the role of RANK signaling in breast cancer cells on the immune cell composition and immunotherapy response of breast tumors. The revised manuscript addresses many of the suggestions and concerns raised by the Referees, and the message of the manuscript is clearly strengthened by the newly added experiments. In particular, the authors have added new experiments to confirm in a more controlled setting that RANK expression in cancer cells influences the immune cell composition of tumors, and they now show more clearly that the synergy between RANK inhibition and immune checkpoint blockade depends on RANK expression in tumors. Newly added in vitro studies reveal that RANK pathway activation in human breast cancer cells enhances the survival and activation of cultured neutrophils. In addition, the authors have performed additional analyses that reinforce the link between the mouse and human studies.

The main conclusion of the authors is that RANK expression in cancer cells promotes an immunosuppressive microenvironment and that targeting RANK pathway may represent a new strategy to enhance an anti-tumor immune response in 'cold' breast cancers.

General comments:

The strength and novelty of this manuscript has been improved by the newly added experiments. The authors have addressed most of my concerns. One of our concerns is not fully addressed, namely the (molecular) mechanisms by which RANK signaling in cancer cells increases TANs and TAMs which restrict T cell immunity. But we appreciate that the authors have strengthened the functional crosstalk between RANK expressing tumor cells, neutrophils and CD8⁺ T cells. We understand that a full dissection of the underlying (soluble) mediators involved in this crosstalk is complex, and not feasible at this stage.

We thank Reviewer #2 for the feedback to improve the manuscript and his understanding of the complexity of including a deeper molecular mechanism within this publication.

There are a few remaining minor points that need (textual) clarification:

Page 27, lines 606-614: the serial transplantation experiment needs more explanation. For the reader, it is not clear what the exact purpose is of the serial transplants. In the rebuttal, the authors refer to their previous publication, however, this should also be clarified in the current manuscript. It is not clear why the authors consider the second transplant as a 'adjuvant' therapy setting, as normally adjuvant therapy implies a treatment after surgical removal of a tumor. But in the experimental set-up, the tumor is implanted in a new mouse, which does not truly mimic a surgical removal setting. The added value of this experiment should be better explained.

The experiment was designed to evaluate whether anti-RANKL treatment would alter the tumor initiating ability of tumor cells (Yoldi et al., Can Res 2016). A limiting dilution assay is required to test the frequency of TICs, which would reflect "recurrence". We agree with the referee that "truly" recurrence (and the effect of adjuvant treatment) should be evaluated after removal of the primary tumor but then, it would be impossible to have a proper TIL quantification. As every experimental setting it can only mimic partially the clinical complexity. In the current manuscript, we avoided the nomenclature neo-adjuvant/adjuvant and only refer to serial passages, as the goal was to determine whether anti-RANKL treatment changes TILs.

Page 28, lines 644-645: the authors state that s100a9 mRNA expression is higher in RANK+/+ tumors, but the statistical analysis does not support this conclusion. The authors should rephrase this. The text has been changed accordingly.

Sup Figure S1a: wrong legends in x- or y- axis of third facs plot (SSC-A is indicated in both axes). Thank you for noticing, the y-axis from the mentioned panel has been corrected to 7AAD staining of dead cells.

Sup Figure S1c: in the legends the authors wrote that each dot represents one tumor transplants derived from 12-13 primary tumors, but in the figure, there are more than 12-13 dots. The authors should correct the sentence or explain the discrepancy.

We apologize for any misunderstanding. Since we are working with tumor transplants derived from different spontaneous primary tumors, we thought it relevant to specify in the figure legend how many individual primary tumors were used as source for the transplants. Each primary tumor will be transplanted into multiple hosts, increasing the total number of tumors analyzed.

Reviewer #3 (Remarks to the Author):

The authors have made suitable changes that address all of my previous concerns. I am also impressed with their extensive and assiduous efforts to address all of the concerns raised by the other 2 reviewers with the inclusion of new data. The point that mouse GEMMs are important models for the purposes used here to overcome many issues of patient tumour heterogeneity is also a key and valid one.

We thank Reviewer #3 for the feedback to improve the manuscript and her/his positive comments.

Reviewer #4 (Remarks to the Author):

The paper covers comprehensive analysis from mice and cell line experiments to human clinical trial. The revised version included additional analyses to improve the paper. Specifically, the control arm in Poetic trial was used to filter out differentially expressed genes due to random sampling differences. Such strategy was justified because both trials have comparable study cohort with majority breast cancer patients in Lum A and B. Bioinformatics analysis was also used to develop RANK metagene and RANKL metagene to compensate IHC limitations of RANK protein detection and to improve selection of tumor-specific pharmacodynamic biomarkers. Analysis results were enhanced by showing high association of RANK metagene with multiple RANK/NFKB pathways and with immuno-response.

Thank you for your appreciation of our work.

A few minor comments:

1. The authors plan to use the ongoing trial, D-BIOMARK (NCT03691311), to validate RANKL inhibition associated immunomodulatory response. However, the clinical trial Gov. website (<https://clinicaltrials.gov/ct2/show/NCT03691311>) indicates change of Ki67 and Cleaved Caspase-3 as the primary endpoints and ratio of RANK/RANKL as secondary endpoint. It is unclear how the D-BIOMARK is capable for validation purpose with sufficient statistical power.

The referee is right; the quantification of TILs is not included as one of primary/secondary endpoints of D-BIOMARK. However, given the relevance of the question and the similar design of D-BIOMARK and D-BEYOND (early breast cancer, 2 doses of single agent denosumab) we will evaluate TILs before and after denosumab treatment. Moreover, RNAseq will also be performed (it is listed as an exploratory endpoint) and CIBERSORT (or similar) analyses will help to confirm putative changes in immune-modulation.

If these results are encouraging our goal is to secure funding for proper characterization of the immunomodulatory role of denosumab in these patients. Moreover, D-BIOMARK includes a control arm, post-menopausal women and TNBC patients. Thus, it will allow to confirm the findings of D-BEYOND in premenopausal BC, but also to extend the observations to post-menopausal and TNBC. It is also a window/exploratory trial (60 patients in total). Based on the findings a new trial focused in the group with the most promising results could be designed to reach statistical power.

2. While plot for comparing RankL metagene between R vs NR was reported in Figure S6, the plot for Rank metagene was missing. Rank metagene was shown to have good correlation with multiple RANK/NFKb pathways. It would be informative to know how many genes in the Rank metagene were overlapped with the correlated pathways. Another question is: Figure S6a had RankL score higher in NR, but Figure 5e had sRankL higher in R group. Any explanation.

The plot comparing RANK metagene between R vs NR was included in Figure 5g. We decided to separate them between a main figure and a supplemental figure because RANK metagene was able to predict the immunomodulatory effect of denosumab while RANKL metagene did not. Of note,

RANK metagene strongly correlated with several public signatures of the RANK and NF-KB pathways in contrast to RANKL metagene.

The RANK/NFKb pathways shared between 1 and 9 genes with the RANK metagene (see table below).

Name	Number of genes shared with RANK metagene
biocarta rankl pathway	1
biocarta nfkb pathway	1
go bone remodeling	4
allmark tnfa signaling via nfkb	9
reactome tnfr2 non canonical nf kb pathway	6
reactome tnf receptor superfamily members mediating non canonical nfkb pathway	2
reactome tnfr2 non canonical nf kb pathway	6
RL 8h UP (WT)	3
RL 8h UP (Tg)	6
RL 24h acini	6

Regarding RANKL metagene score and sRANKL levels, the differences derive from the sample source. RANKL metagene would measure mRNA RANKL levels (and the associated 100 genes) in the tumor itself (i.e. how much RANKL is produced by tumor cells or tumor stroma). In contrast, sRANKL measures the amount of protein circulating in the blood stream, therefore including sources of RANKL other than the tumor.

3. It would be more informative to report immune-response in the Table 1 of clinicopathological features of the 24 evaluable patients.

Table 1 has been modified as requested:

N		24
Interval surgery-Denosumab	Median days (range)	13 (9-21)
Age	Median years (range)	44 (35-51)
Size	> 2cm	11 (45.8%)
Nodal status	Positive	4 (16.7%)
Histological grade	High	8 (33.3%)
Molecular subtypes	LumA	10 (41.7%)
	LumB	9 (37.5%)
	HER2	4 (16.7%)
	TNBC	1 (4.2%)

Immune response*	Percentage of patients	11 (45.8%)
------------------	------------------------	------------

* $\geq 10\%$ increase in TIL infiltration

4. The study showed rank metagene as a potential biological endpoint. It is unclear if it is associated with clinical endpoints.

The computation of RANK metagene was not originally included in the clinical trial protocol. As the referee highlights RANK metagene was generated to compensate IHC limitations of RANK protein detection and to reflect activation of RANK signaling pathway in the tumors at baseline. We found that RANK metagene in the tumors at baseline could predict the immunomodulatory effect of denosumab. Thus, it can be considered a predictive biomarker for the selection of patients that may benefit from denosumab, but not as a biological endpoint.

REVIEWERS' COMMENTS

Reviewer #4 (Remarks to the Author):

Reading opposite result of RankL score higher in NR (Figure S6a) and sRankL higher in R group (Figure 5e), the authors consider the difference is due to sample source variation and indicate "sRANKL measures the amount of protein circulating in the blood stream, therefore including sources of RANKL other than the tumor". Does it mean sRNAK result is due to noise and why such noise convert to significant opposite result? What is the purpose to include the sRNAK result?

Point by point response to referee #4

October 12th

REVIEWERS' COMMENTS

Reviewer #4 (Remarks to the Author):

Reading opposite result of RankL score higher in NR (Figure S6a) and sRankL higher in R group (Figure 5e), **the authors consider the difference is due to sample source variation and indicate** "sRANKL measures the amount of protein circulating in the blood stream, therefore including sources of RANKL other than the tumor". Does it mean sRNAK result **is due to noise** and why **such noise convert to significant opposite result**? What is the purpose to include the sRNAK result?

We understand that when the referee says sRNAK, she/he means sRANKL (soluble RANKL), as shown in Figure 5e.

The IHC results from Figure S6a would most likely reflect the membrane RANKL isoform, expressed in tumor or stromal cells. It is unlikely that soluble RANKL infiltrating the tumor from the blood stream would be detected by IHC, unless "immobilized" by binding its receptor on a cell membrane. Therefore, sRANKL from Figure 5e detected by ELISA and tumor (membrane) RANKL from Figure S6a are two different parameters, which we would not expect to be necessarily linked.

The question of the referee is not totally clear to us; we do not understand what she/he means by "noise" or "sample source variation". The fact that at baseline higher levels of soluble RANKL and RANK-metagene in the tumor, but not RANKL expression in the tumor, are higher in "responders", suggest that systemic RANKL is responsible of RANK pathway in the tumor (RANK metagene), on one hand, and the denosumab-driven immune effect is due to the blockage of soluble RANKL.